# *Arabidopsis* cryptochrome 2 forms photobodies with TCP22 under blue light and regulates the circadian clock

Weiliang Mo[1,3], Junchuan Zhang[1,3], Li Zhang[1,3], Zhenming Yang[1,3], Liang Yang[2], Nan Yao[2], Yong Xiao[1], Tianhong Li[1], Yaxing Li[2], Guangmei Zhang[2], Mingdi Bian[1], Xinglin Du[1] & Zecheng Zuo[1,2 ✉]

Cryptochromes are blue light receptors that regulate plant growth and development. They also act as the core components of the central clock oscillator in animals. Although plant cryptochromes have been reported to regulate the circadian clock in blue light, how they do so is unclear. Here we show that *Arabidopsis* cryptochrome 2 (CRY2) forms photobodies with the TCP22 transcription factor in response to blue light in plant cells. We provide evidence that PPK kinases influence the characteristics of these photobodies and that together these components, along with LWD transcriptional regulators, can positively regulate the expression of *CCA1* encoding a central component of the circadian oscillator.

[1] Jilin Province Engineering Laboratory of Plant Genetic Improvement, College of Plant Science, Jilin University, 5333 Xi'an Road, Changchun 130062, China. [2] Basic Forestry and Proteomics Research Center, Fujian Agriculture and Forestry University, Fuzhou 350002, China. [3] These authors contributed equally: Weiliang Mo, Junchuan Zhang, Li Zhang, Zhenming Yang. ✉email: zuozhecheng@jlu.edu.cn

Light is not only an energy source of photosynthetic organisms but an important environmental signal for most living creatures[1,2]. The environmental cue provided by light is crucial for organisms to synchronize their intrinsic circadian clock with the 24 h solar day on earth, allowing them to anticipate environmental changes, make behavioral decisions, and to impact their fitness and survival[3–5]. Plants sense light through multiple photoreceptors[6–8]. Two decades ago, it was reported that blue light receptor cryptochromes mediate blue light input to the circadian clock in *Arabidopsis*, however, the mechanism remains unknown[9,10]. Cryptochromes are conserved in most organisms[11,12]. In contrast to animal cryptochromes, *Arabidopsis* cryptochromes are typically considered as photoreceptors rather than direct regulators or components of the clock central oscillator[13,14].

In previous studies, we searched for proteins that interact with *Arabidopsis* CRY2 to transduce signals in response to blue light. Dozens of cryptochrome-interacting proteins have been identified in the blue light signaling pathway, including SUPPRESSOR OF PHYTOCHROME As (SPAs), BLUE-LIGHT INHIBITOR OFCRYPTOCHROMEs (BICs), and PHOTOREGULATORY PROTEIN KINASES (PPKs)[15–17]. However, none of the circadian clock proteins have been found in CRY2-interacting protein pools, and it remains unclear how CRY2 transduces the blue light signal into the circadian clock. Recently, we utilized high-resolution mass spectrometry (Orbitrap Fusion Lumos) to further investigate CRY2-interacting proteins, and identified a circadian clock protein TEOSINTE BRANCHED1-CYCLOIDEA-PCF 22 (TCP22) that interacted with CRY2. However, TCP22 exhibited a constitutive physical interaction with CRY2, as opposed to other CRY2 interacting proteins that interact with cryptochromes in a blue light-dependent manner. Since the CRY2-TCP22 interaction lacked a light response, our further investigation of the mechanism on CRY2-mediated blue light input to the circadian clock was impeded.

Inspired by the blue-light-specific formation of CRY2 photobodies, as reported in our previous research[15,18], we re-assessed the molecular relationship between CRY2 and TCP22 in this study. We found that while the protein–protein interaction is constitutive, CRY2 and TCP22 form photobodies in a blue-light-dependent manner. We also show that the characteristics of CRY2 containing photobodies can be regulated by PPK kinases and the TBS motif of the CIRCADIAN CLOCK ASSOCIATED 1 (*CCA1*) promoter and provide evidence that CRY2 and TCP22 may regulate *CCA1* expression via LIGHT REGULATED WDs (LWDs).

## Results

### CRY2 forms photobodies with TCP22 in a blue light-dependent manner.
We first sought to test the mass spec result that CRY2 physically interacts with TCP22 consistently in both the dark and blue light (Fig. 1a). Yeast two-hybrid and co-immunoprecipitation assays suggested that the physical interaction of CRY2 and TCP22 was nearly constant (Fig. 1b, c). BiFC assays indicated that in darkness, nYFP-CRY2 interacted with cYFP-TCP22 to reconstitute a fluorescent signal (Supplementary Fig. 1a). On the other hand, in response to blue light, the interaction between nYFP-CRY2 and cYFP-TCP22 not only generated the fluorescent signal but further formed fluorescent photobodies (Supplementary Fig. 1a). The blue light specificity of CRY2-TCP22 photobody formation was assayed by multi-spectroscopic analyses (Supplementary Fig. 2). As the split fluorescence protein tags used in BiFC assays could also absorb light and potentially affect photobody formation, we also tested CRY2-TCP22 photobody formation in response to blue light utilizing an immunostaining assay without fluorescent protein tags. As shown in Fig. 1d, e, a similar amount of Myc-TCP22 formed photobodies with the native CRY2 only after sufficient blue light irradiation (30 or 60 min) in *Myc-TCP22* over-expressing plants. In addition, the photobodies of CRY2-TCP22 presented a larger size after a longer blue light irradiation, which suggests the CRY2-TCP22 photobodies are dynamic in plant cells (Fig. 1f). Since the over-expression of TCP22 may promote photobody formation by affecting protein concentration, we then transformed the *Myc-TCP22* with its native promoter into the *tcp20tcp22* mutant and investigated photobody formation of CRY2-TCP22 in *ProTCP22::MycTCP22/tcp20tcp22* plants. As above, TCP22 formed photobodies with the native CRY2 only under blue light in *ProTCP22::MycTCP22/tcp20tcp22* plants, compared with that under red light (Fig. 1g–i). Taken together, these results suggest that blue light not only affects CRY2 protein–protein interactions, but also induces photobody formation with TCP22.

### TCP22 photobody formation is dependent on CRY2 and regulates the properties of CRY2 condensates.
To understand the biological process of CRY2-TCP22 photobody formation, we first examined the photobody properties of CRY2 and TCP22 proteins, respectively. As we reported previously, CRY2-GFP could form photobodies in response to blue light in plant cells (Supplementary Fig. 1b)[19]. Endogenous CRY2 can generate photobodies after blue light irradiation in wild-type seedlings as observed via an immunostaining assay (Fig. 2a, b). However, no photobody was observed in *cry2* protoplasts expressing TCP22, even after blue light irradiation (Supplementary Fig. 1c, upper panel), unless TCP22 co-expressed with CRY2 (Supplementary Fig. 1c, bottom panel). Likewise in seedlings immunostaining assays indicated that over-expression of TCP22 is not sufficient to support photobody formation in *cry1cry2* mutant, whereas TCP22-CRY2 photobody formed in the Col-4 background (Fig. 2c–e). Likewise, TCP22 driven by its own promoter (*proTCP22*) only formed photobodies with CRY2 in blue light, whereas *proTCP22::Myc-TCP22* did not form photobodies in the *cry1cry2* background or in red light (Supplementary Fig. 3a). These results suggested that TCP22 forms photobodies in a blue light and CRY2-dependent manner.

To further investigate the biochemical properties of the TCP22 photobody formation, we examined such process with CRY2 and CRY2$^{D387A}$, a chromophore-deficient mutant in which the aspartic acid residue was changed to alanine at position 387. Since the residue D387 of CRY2 is part of the FAD-binding pocket absorbing blue light, CRY2$^{D387A}$ does not contain the flavin and is "blind" to blue light[20]. TCP22 formed photobodies in protoplasts with CRY2 (Supplementary Fig. 3b, upper panel), but no photobodies were formed in protoplasts expressing TCP22 and CRY2$^{D387A}$ (Supplementary Fig. 3b, middle panel). This suggests that the formation of TCP22 photobodies occurs in a CRY2 dependent manner; in addition, the formation of TCP22 photobodies depends on the chromophore (flavin) of CRY2.

Interestingly, both CRY2 only and CRY2-TCP22 photobodies were bigger after longer blue light irradiation, but the CRY2-only photobodies (Supplementary Fig. 3b, bottom panel) had a slower formation rate and a smaller size compared with the CRY2-TCP22 photobodies (Supplementary Fig. 3b, upper panel) at the same blue-light-radiation time point (Supplementary Fig. 3b–d), which was verified by the immunostaining assay. Even with a similar amount of CRY2 (Supplementary Fig. 4b), the CRY2-TCP22 photobodies grew faster than CRY2 only photobodies after an equal blue light irradiation time (Supplementary Fig. 4a, c). Taken together, these results suggest that the formation of TCP22 photobodies is dependent on CRY2, whereas TCP22 further promotes photobody size under blue light irradiation.

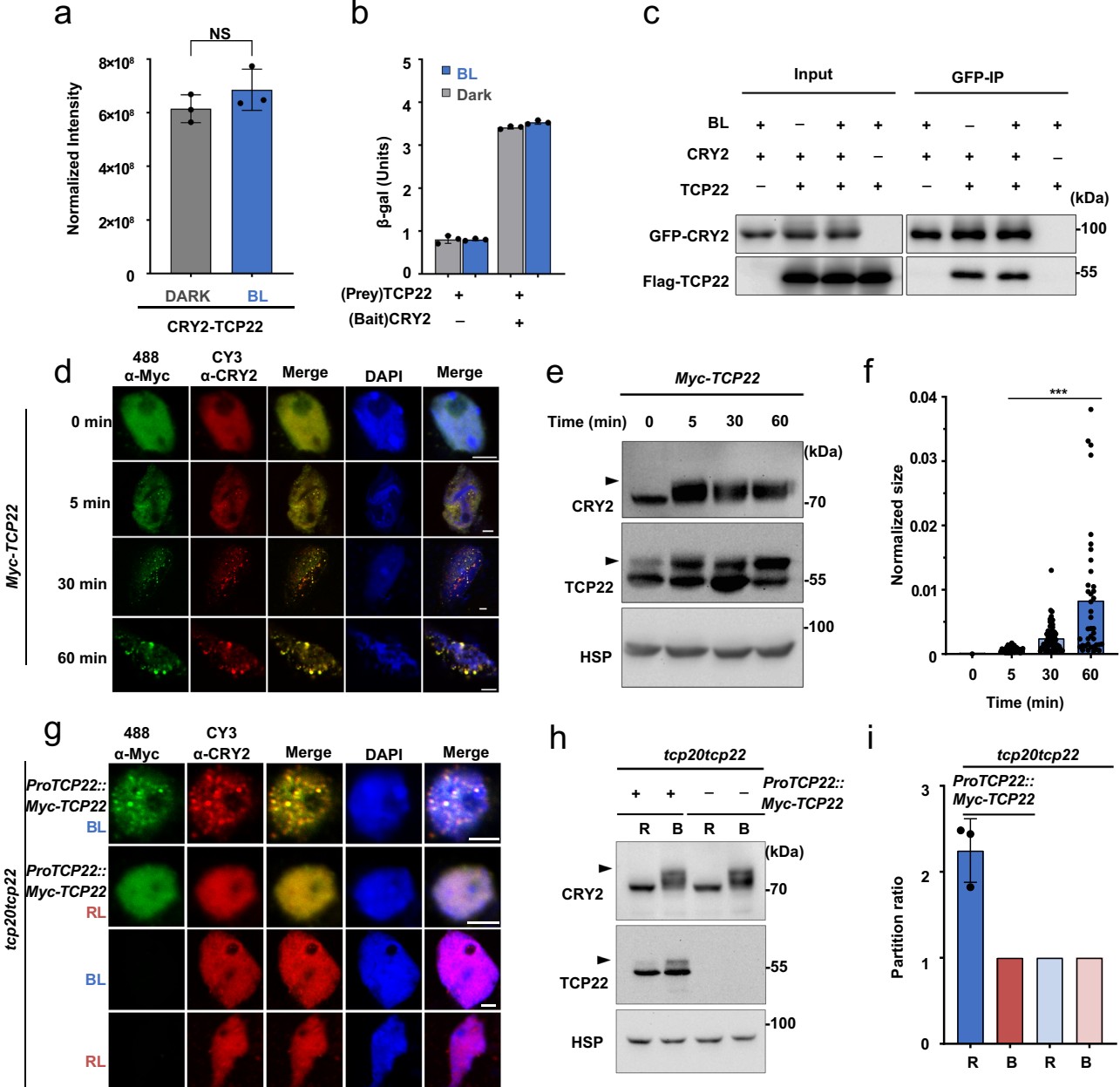

**Fig. 1 CRY2 and TCP22 exhibit blue-light-independent interaction but blue-light-dependent photobody formation. a** Mass Spectrometry analysis showing the interaction of CRY2 and TCP22 in vivo. Normalized intensity of three biological replicates from dark and blue light (BL, 30 μmol m$^{-2}$ s$^{-1}$) treated seedlings, respectively. Two-tailed student's t-test was used for statistical tests. Data are presented as mean ± SD ($n = 3$ independent experiments). NS, No significance, $p = 0.257$. **b** β-galactosidase activity of yeast expressing TCP22 (prey) and CRY2 (bait) or empty vector. Data are presented as mean ± SD ($n = 3$ technical repeats). **c** Co-immunoprecipitation (Co-IP) assay showed the interaction of CRY2 and TCP22 in HEK-293T cells. The cells were treated with blue light (+ BL, 30 μmol m$^{-2}$ s$^{-1}$) for 3 h or kept in the dark (−BL). The immunoprecipitation signals were probed by anti-GFP (CRY2) or anti-Flag (TCP22), respectively. Two independent experiments show similar results. **d** Photobody formation of CRY2/TCP22 in the nuclei detected by immunostaining assay in *Myc-TCP22*/Col4 plants irritated by blue light (30 μmol m$^{-2}$ s$^{-1}$). Scale bar, 3 μm. **e** Immunoblots of the samples prepared in (**d**). The arrowhead indicated the phosphorylated TCP22 or CRY2. **f** Normalized size analysis of (**d**) ($n = 5$ nuclei under blue light for 5 min and $n = 4$ nuclei under blue light for other times), Two-tailed student's t test was used for statistical tests(***$p < 0.001$, $p = 4.44 \times 10^{-5}$, $1.11 \times 10^{-5}$, $1.49 \times 10^{-4}$ respectively). **g** Immunostaining of nucleus showed the CRY2/TCP22 photobodies formed endogenously in *ProTCP22::TCP22/tcp20tcp22* seedlings. Scale bar, 3 μm. **h** Immunoblots of samples prepared in (**g**). The arrowhead indicated phosphorylated TCP22 or CRY2. **i** Partition ratio analysis of (**g**). Data are presented as mean ± SD ($n = 3$ nuclei).

We next analyzed the dynamic properties of these photobodies and found that CRY2 and CRY2-TCP22 photobodies could fuse with each other (Supplementary Fig. 5a, b), indicating the CRY2 and CRY2-TCP22 photobodies are dynamic in plant cells. Similar to liquid-liquid phase separation condensates, photobody formation was sensitive to 1, 6-hexanediol, a chemical that disrupts

hydrophobic interactions and induces phase separation assemblies (Supplementary Fig. 5c, d).

**The TBS motif of *CCA1* affects the properties of CRY2-TCP22 condensates.** Since CRY2 and TCP22 formed photobodies under

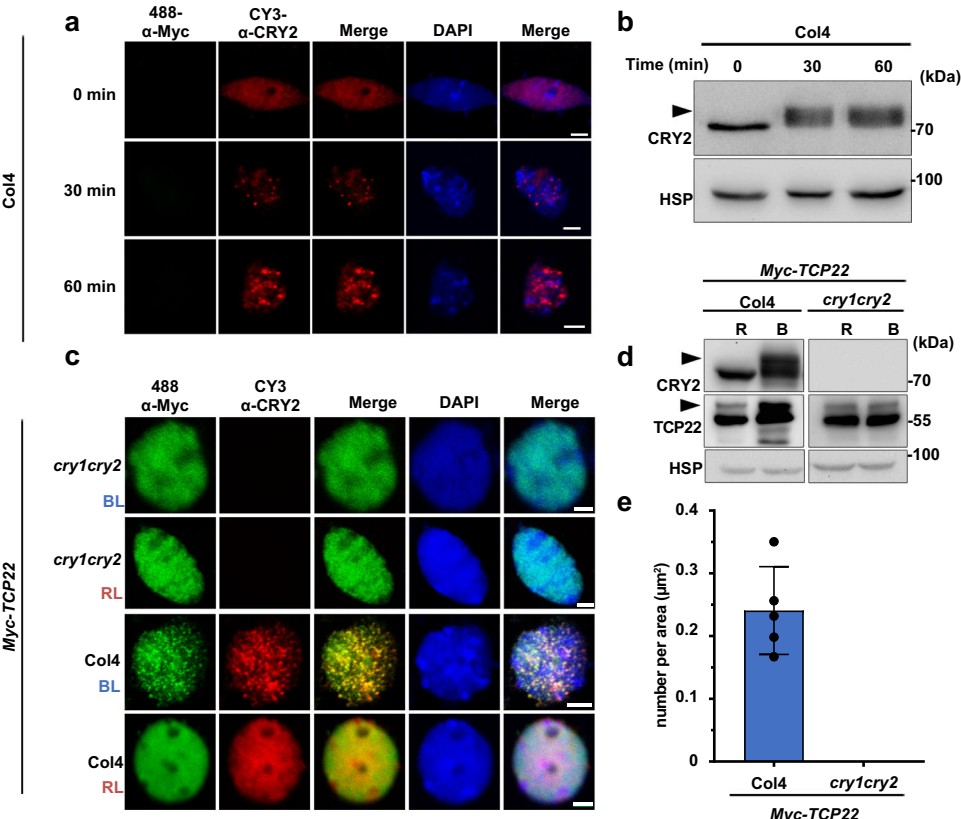

**Fig. 2 TCP22 forms photobodies in blue-light in a CRY2-dependent manner. a** Immunostaining of nucleus showing the CRY2 photobodies formed endogenously in *Arabidopsis* seedlings. Scale bar, 3 μm. Three independent experiments showed similar results. Immunoblots of samples prepared in (**b**). The arrowhead indicated the phosphorylated CRY2. **c** Immunostaining of nucleus showing the endogenous photobody formation in *Myc-TCP22*/Col4 and *Myc-TCP22/cry1cry2* seedlings. Scale bar, 3 μm. **d** Immunoblots of samples prepared in (**c**). The arrowhead indicated the phosphorylated TCP22 or CRY2. **e** Statistical analysis for the number of photobodies per area in (**c**). Data are presented as mean ± SD ($n = 5$ nuclei).

blue light, we next wished to investigate whether CRY2 inputs the blue light signal to the circadian clock via TCP22 and to assess the contribution of the CRY2-TCP22 photobodies in this process. As it was previously observed that TCP22 activates the *CCA1* promoter in white light[21], we first investigated whether CRY2 mediates blue light input to the circadian clock and regulates the rhythm of the circadian clock via TCP22. In blue light, the transcript level of *CCA1* was increased in plants overexpressing CRY2 or TCP22 and reduced in the *cry1cry2* or *tcp20tcp22* double mutants, compared with the wild type (Fig. 3a). In addition, the activator activity of the overexpressed TCP22 was significantly reduced in *cry1cry2* background (*Myc-TCP22/cry1cry2*) when compared with either wild-type or plants overexpressing TCP22 in a wild type background (Fig. 3b). This indicates that both CRY2 and TCP22 promote the transcription of *CCA1* under blue light, and that the activity of TCP22 in promoting *CCA1* expression is largely dependent on crytophrome. The transcription of *CCA1* maintained a robust rhythm under red light (Supplementary Fig. 6a), but *CCA1* became arrhythmic in the *cry1cry2* background under blue light, irrespective of whether TCP22 was overexpressed or not (Fig. 3c and Supplementary Fig. 6b). A blue light pulse assay showed that TCP22 significantly enhanced the amplitude of the *CCA1* expression and altered the rhythm of the central oscillator after blue light pulses (Fig. 3d). However, the activity of TCP22 was suppressed in the *cry1cry2* background (Supplementary Fig. 6c–f), suggesting that TCP22 regulated *CCA1* rhythm via the enhancement of *CCA1* transcript level, with transcriptional activator activity being dependent on cryptochrome under blue light.

We next investigated the contribution of CRY2-TCP22 photobody formation in promoting *CCA1* transcription. It was reported recently that the concentration of transcription factors and *cis*-elements of the regulated gene in phase separated condensates can enhance transcriptional output and that condensate lifetime and transcription are positively correlated[22,23]. We examined whether CRY2-TCP22 photobody formation was impacted by the DNA elements of *CCA1*. As shown in Fig. 3e, CRY2 and TCP22 formed condensates in vitro at a lower protein concentration when the TBS motif[21,24] of the *CCA1* promoter region was present, compared to a mutated TBS motif or a random piece of DNA. This suggests that CRY2-TCP22 photobodies could recruit DNA fragments in a sequence-dependent manner. Intriguingly, in the presence of the TBS motif, the threshold concentration for CRY2-TCP22 condensate formation in vitro was lower and the number of CRY2-TCP22 droplets was increased, compared to samples without DNA, suggesting the DNA fragment of *CCA1* promoter could accelerate the formation of CRY2-TCP22 photobodies (Supplementary Fig. 6g–i). Together with the observation that both CRY2 and TCP22 could bind to the *CCA1* promoter (Fig. 3f), these results are consistent with a model where CRY2-TCP22 photobodies may enhance the CRY2-TCP22 protein concentration around the TBS motif of the *CCA1* promoter. Nevertheless, further work would be needed to test whether the observed CRY2-TCP22 photobodies colocalize with the TBS motif in planta.

**The properties of CRY2-TCP22 photobodies are regulated by PPK1 and phosphorylation of TCP22.** To further investigate

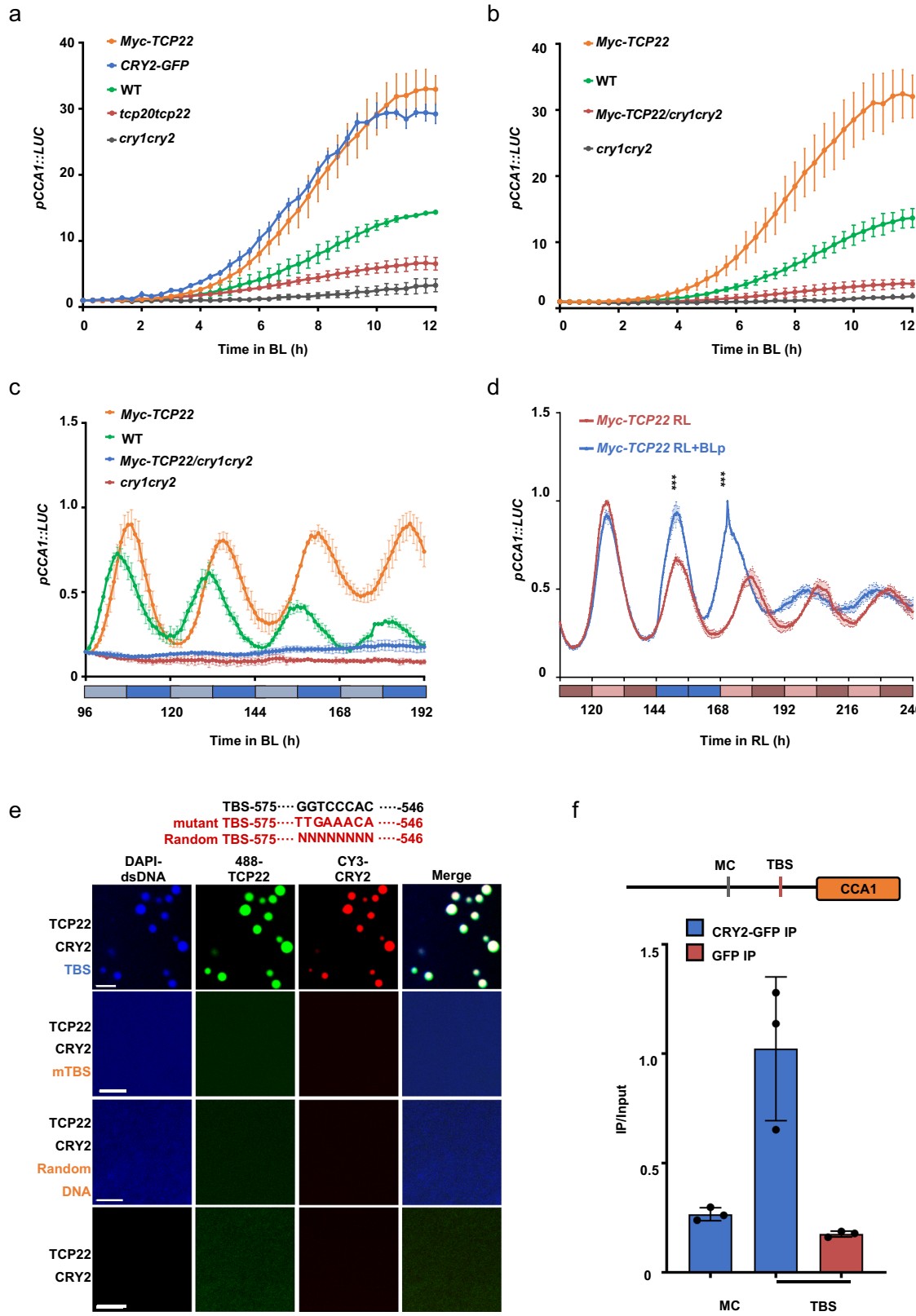

CRY2–TCP22 photobodies, we assessed the biochemical properties of CRY2 and TCP22 after blue light irradiation. TCP22 was phosphorylated in plant cells (Supplementary Fig. 7a). Similar to CRY2, as previously reported[16,25], the phosphorylation of TCP22 was enhanced by blue light in a CRY2-dependent manner (Fig. 4a). To screen the kinase which phosphorlated TCP22, we

identifed kinases PPK1 interacting with TCP22 through co-immunoprecipitation assay. Besides PPK1, PPK2, PPK3 and PPK4[16] were previously reported to interact with CRY2. Via yeast two hybrid and expression in HEK293T cells and protoplasts we found that TCP22 could interact with PPK1–4 (Supplementary Fig. 7b–d and Supplementary Fig. 8a–c). PPKs could phosphorylate

**Fig. 3 CRY2-TCP22 photobody formation is influence by the TBS motif of *CCA1*. a** Bioluminescence analysis of *pCCA1::LUC* expression in indicated genotypes under blue light (BL, 10 μmol m$^{-2}$ s$^{-1}$). Several seedlings of different genotypes were grown on 1/2 MS solid medium in the dark for 5 d, then transferred to blue light (10 μmol m$^{-2}$ s$^{-1}$). Data are presented as mean ± SD ($n = 3$ independent experiments). **b** Bioluminescence analysis of *pCCA1::LUC* expression in indicated genotypes under blue light (BL, 10 μmol m$^{-2}$ s$^{-1}$). The conditions were the same as (**a**). Data are presented as mean ± SD ($n = 3$ independent experiments). **c** Bioluminescence analysis of *pCCA1::LUC* expression in indicated genotypes. Seedlings were grown on 1/2 MS solid medium in 12 h dark/12 h light (75 μmol m$^{-2}$ s$^{-1}$ white light) treatment for 3 days and then transferred into continuous blue light (BL, 5 μmol m$^{-2}$ s$^{-1}$). Data are presented as mean ± SD ($n = 3$ independent experiments). **d** Blue light pulse assay in *Myc-TCP22*/WT. 12 h light/12 h dark grown plants were transferred to continuous red light (RL, 10 μmol m$^{-2}$ s$^{-1}$), then treated with or without a blue light pulse at the indicated time point (BLp, 15 μmol m$^{-2}$ s$^{-1}$). Data are presented as mean ± SD ($n = 3$ independent experiments). Two-tailed student's *t* test: ***$p < 0.001$, $p = 8.68 \times 10^{-4}$, $5.96 \times 10^{-5}$ respectively. **e** Mutant TBS or random DNA failed to enhance the formation of CRY2-TCP22 photobodies in vitro. 6 μM iFlour-488-labeled TCP22 and 1 μM CY3-labeled CRY2 were mixed with or without 25 μM DAPI labeled 30 bp dsDNA containing TBS, mutant TBS or random DNA. Then the images were captured under 405, 488, and 550 nm lasers. Similar results were observed in three independent repeats. TBS: TCP22 binding site at *CCA1* promoter. **f** CRY2 associated with the TBS-containing region of *CCA1* promoter in vivo. ChIP assays involved the use of the anti-GFP with CRY2-GFP or GFP transgenic line. The input and GFP-IP product were analyzed by qPCR. Data are presented as means ± SD ($n = 3$ technical repeats). MC: motif of control.

TCP22 in heterologous HEK293T cells (Fig. 4b and Supplementary Fig. 8d–f) and *Arabidopsis* seedlings (Supplementary Fig. 8g). Mass spectrometry analysis showed that all the phosphorylation sites of TCP22 regulated by PPK1 were concentrated in a short region (37AA–50AA) near the N terminal, and the abundance of those phosphorylation sites was enhanced by the presence of PPK1 but not the mutant (PPK1$^{D267N}$)[16] (Fig. 4c). We thus examined the relationship between these three proteins (CRY2, TCP22 and PPK1). As shown in Fig. 4d and Supplementary Fig. 7e, when expressed in HEK cells, CRY2 increased the phosphorylation of TCP22 and the protein–protein-interaction between PPK1 and TCP22 in blue light, in contrast to the CRY2$^{D387A}$ or PPK1$^{D267N}$ mutants. Split-luciferase assays also suggested that CRY2 promoted the interaction of nLUC-TCP22 with cLUC-PPK1 in plant cells under blue light (Fig. 4e). These results are consistent with our observation that the phosphorylation of TCP22 depended on CRY2 and blue light. We noticed that, although TCP22 could interact with PPK1 and be slightly phosphorylated, no TCP22-PPK1 condensate was observed (Supplementary Fig. 7d) unless CRY2 was co-expressed (Fig. 4f), which indicates that TCP22-PPK1 condensate formation takes place in a CRY2-dependent manner. It is possible that CRY2 not only enhances the physical interaction of TCP22-PPK1, but also recruits TCP22 and PPK1 to form a blue light induced condensate in plant cells, which leads to the complete phosphorylation of TCP22.

We next asked whether the recruitment of PPK1, or the phosphorylation of CRY2 and TCP22 plays a role in the maintenance of CRY2-TCP22 photobodies. As indicated in Fig. 4i, CRY2-TCP22 photobodies were slightly more dynamic when co-expressing with PPK1 after blue light irradiation. It was recently reported that after a prolonged incubation or after a few cycles of assembly and disassembly, LLPS condensates usually shift to an irreversible aggregating stage[26]. We thus examined the properties of CRY2-TCP22-PPK1 photobodies under sequential cycles of blue light and dark conditions. When PPK1 was co-expressed in plant cells, CRY2-TCP22-PPK1 condensates assembled after 10 min of blue light irradiation and fully disassembled each time after 10 min of darkness incubation (Fig. 4g, upper panel). These photobodies even maintained a robust dynamism of assembly and disassembly after several blue-dark cycles (Fig. 4g, h, j). By contrast, without PPK1, the CRY2-TCP22 photobody dissolution was reduced even after the first blue-dark cycle (Fig. 4g, bottom panel) and the photobodies were significantly less dynamic after the second cycle (Fig. 4j). Taken together, these results suggest that PPK1 maintains the dynamicity of CRY2-TCP22 photobodies preventing the formation of irreversible aggregates after the cycles of assembly and disassembly; while, CRY2 enhances the interaction of TCP22-PPK1 and the phosphorylation of TCP22 under blue light. In

addition, PPK1 also enhanced the expression of *CCA1* (Fig. 4k) and controlled the activity of TCP22 to regulate the *CCA1* circadian rhythm under blue light (Fig. 4l, m).

To explore how PPK1 mediated phosphorylation of TCP22 might regulate the properties of the CRY2-TCP22 photobodies and affect the expression and the circadian rhythm of *CCA1*, we analyzed the properties of photobodies produced by a serine (threonine)-to-alanine mutant of TCP22 (mTCP22$^{12STA}$), in which twelve serine or threonine residues of TCP22 phosphorylated by PPK1 were mutated to alanine to mimic non-phosphorylated TCP22. Compared with the wild type TCP22 of equal amount, the mutant TCP22$^{12STA}$ was not phosphorylated by PPK1 (Supplementary Fig. 9a) and PPK1 had no activity in promoting condensate formation of mutant TCP22$^{12STA}$ droplets in vitro (Supplementary Fig. 9b–d). Likewise, in plant cells mTCP22$^{12STA}$ neither enhanced the dynamics nor improved the formation of CRY2 photobodies (Supplementary Fig. 9e–i) or activated the expression of *CCA1* (Supplementary Fig. 9j–l). In contrast, the overexpression of mTCP22$^{12STA}$ showed a dominant negative effect to inhibit the activity of TCP22, CRY2 and PPK1 in plant cells (Supplementary Fig. 9j–l). This suggests that PPK1 could enhance the formation and dynamics of CRY2-TCP22 photobodies to promote the expression of *CCA1* and regulate the circadian clock via the phosphorylation of TCP22.

**CRY2 coordinates with TCP22, LWD1, and PPK1 to mediate blue light input to the central oscillator**. To further assess how CRY2-TCP22-PPK1 photobodies may mediate blue light input to the clock, we first reconstituted the "blue light input to the clock" signaling pathway in the mammalian cell system as we previously described[27,28], integrating blue light perception, downstream protein modification and the activation of central oscillator component *CCA1* successively. Surprisingly, neither TCP22 nor the TCP22-PPK1 complex enhanced the expression of *CCA1* in our reconstituted system (Supplementary Fig. 10a) and CRY2 can't promote the transcription activity of TCP22 (Supplementary Fig. 10b), which are inconsistent with the results we obtained in plant cells. This implies that other proteins may also be involved in this machinery.

Previous studies have reported that TCP22 collaborates with the transcription co-factors LWDs[21,29,30], suggesting LWDs may also regulate blue light input to the circadian clock. We, therefore, investigated whether LWDs can interact with CRY2, TCP22, and PPK1 and form photobodies with the above proteins. LWDs interacted with CRY2, PPK1, and TCP22 in heterologous HEK293T cells (Supplementary Fig. 11a, b and Fig. 5a). Notably, LWD1 exhibited a stronger protein–protein interaction with the phosphorylated TCP22, compared with the un-phosphorylated TCP22 (Fig. 5a, b). This suggests that PPK1 enhances the LWD-

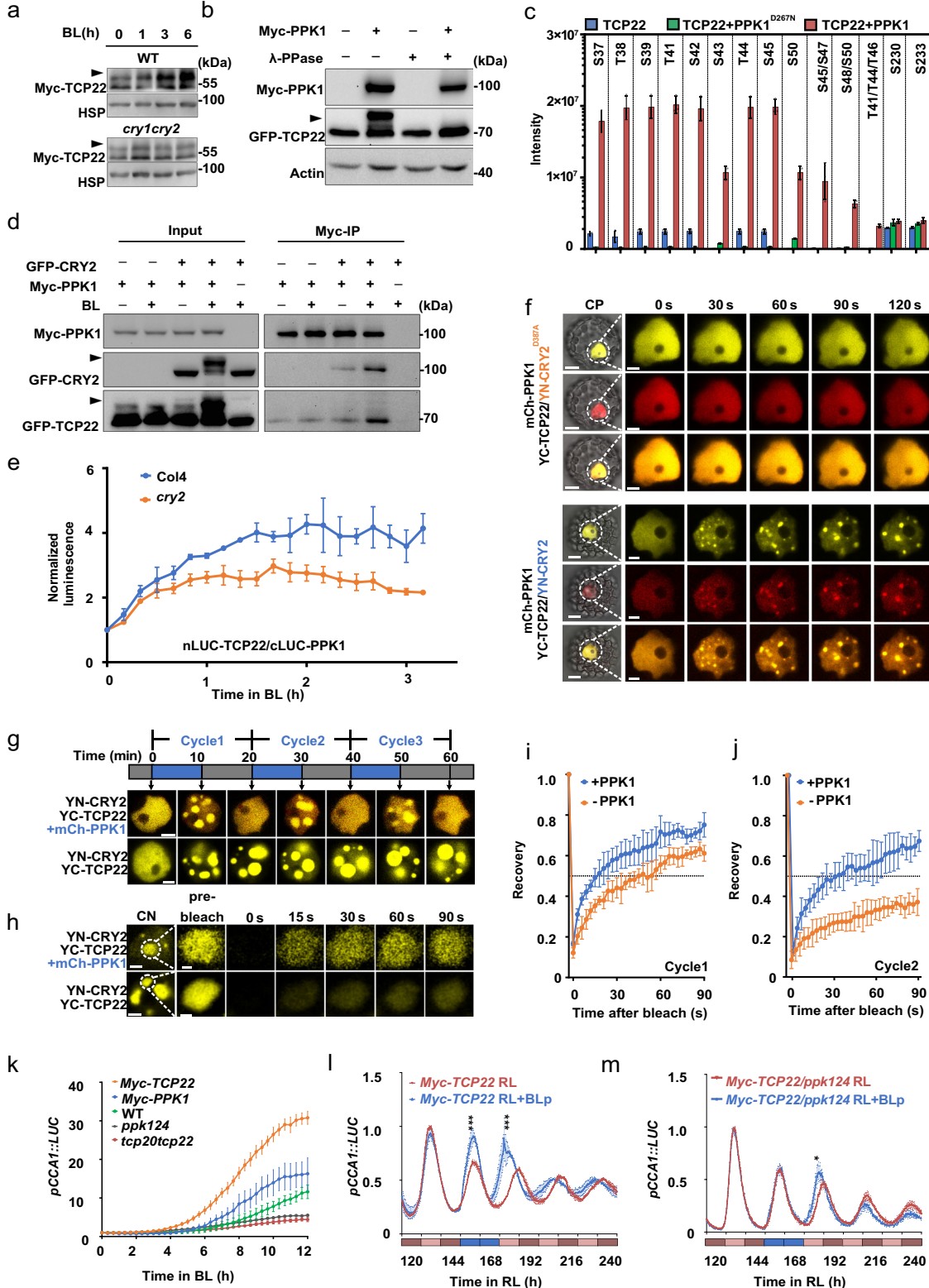

TCP22 interaction. Furthermore, the interaction of LWD-TCP22 could be further enhanced by CRY2 after blue light irradiation (Fig. 5c). LWD2 (the homolog of LWD1) displayed functional similarities with LWD1 (Supplementary Fig. 11c–e). Next, we examined whether CRY2 could recruit LWDs into the photobodies. As shown in Fig. 5d, both LWD1 and LWD2 formed the photobodies with CRY2 in blue light when expressed in protoplasts, but the partition ratio and the circularity of CRY2-

LWD1 condensates were significantly lower than CRY2-TCP22 condensates, and even lower than the condensates of CRY2-only (Supplementary Fig. 12a, b). When CRY2, TCP22 and LWD1 were co-expressed together, these photobodies exhibited a more spherical shape in plant cells (Fig. 5e), compared with CRY2-LWD1 photobodies (Fig. 5d, upper panel). In addition, PPK1 significantly increased the size and partition ratio of CRY2-TCP22-LWD1 condensates both in vivo (Fig. 5e–g) and

**Fig. 4 PPK1 can phosphorylate CRY2 and TCP22 and regulate the properties of CRY2-TCP22 photobodies. a** Immunoblots showing the phosphorylation of TCP22 in indicated genotypes under blue light (45 μmol m$^{-2}$ s$^{-1}$) for the indicated time. The arrowhead indicated the phosphorylated TCP22, same in the below. Two independent experiments showed similar results. **b** Dephosphorylation assay of TCP22 in HEK-293T cells by λ-PPase. Actin was used as a loading control. Two independent experiments showed similar results. **c** Mass Spectrometry analysis of the phosphosites of TCP22 phosphorylated by PPK1 or PPK1$^{D267N}$. Data are presented as mean ± SD ($n = 3$ independent experiments). **d** Co-IP assay showing the interaction of CRY2, TCP22, and PPK1 in HEK-293T cells. **e** Split luciferase assay in WT and *cry2* under blue light (10 μmol m$^{-2}$ s$^{-1}$). Data are presented as mean ± SD ($n = 3$ independent experiments). **f** Photobody formation of PPK1, CRY2 (CRY2$^{D387A}$), and TCP22 in *Arabidopsis* (Col4) protoplasts. Scale bar, 5 μm for the CP (Complete protoplast), 3 μm for nucleus. Three independent experiments showed similar results. **g** Assembly and disassembly of the photobodies in Col4 protoplasts. A time cycle of 10 min on/10 min off was used to treat protoplasts for assembly and disassembly of the photobodies. Scale bar, 3 μm. Three independent experiments showed similar results. **h** FRAP assay of photobodies of (**g**). Scale bar, 3 μm (for CN); 0.5 μm (for photobodies). Three independent experiments showed similar results. **i, j** Quantification of FRAP. **i** FRAP after the first cycle, **j** FRAP after the second cycle, data are presented as mean ± SD ($n = 5$ nuclei). **k** Bioluminescence analysis of *pCCA1::LUC* expression in the indicated genotypes. The condition was described above in Fig. 3a. Data are presented as mean ± SD ($n = 3$ independent experiments). Blue light pulse assay of *pCCA1::LUC* in *Myc-TCP22*/WT (**l**), and in *Myc-TCP22/ppk124* (**m**). Data are presented as mean ± SD ($n = 3$ independent experiments). The condition was described above in Fig. 3d. Two-tailed student's *t* test: *or *** represents *p* value < 0.05 or < 0.001, respectively. *p* = $5.12 \times 10^{-4}$,$1.83 \times 10^{-5}$ in (**l**), *p* = 0.02 in (**m**).

in vitro (Fig. 5h and Supplementary Fig. 12c, d). These observations are consistent with previous studies[31–33], suggesting that larger-condensates are able to include more transcription factors.

According to a recent report, transcription is enhanced by the co-condensation of transcriptional regulators in liquid condensates[34] and that condensate formation of transcription factors increase gene expression[35,36]. In this study, we also found CRY2, TCP22, PPK1 and LWD1 could co-locate in a photobody when expressed in protoplasts. We, therefore, investigated how those four proteins (CRY2, TCP22, PPK1 and LWD1) influence the transcription of *CCA1*. We co-expressed TCP22 with LWD1 in the transcription reconstitution system described above. The expression of the *CCA1*-promoter-driven Luciferase was up-regulated when both TCP22and LWD1 were expressed in the presence of PPK1 compared to TCP22 or LWD1 and PPK1 alone (Fig. 5i). *CCA1* expression level was higher when by PPK1 (Fig. 5i). A non-phosphorylated mTCP22$^{12STA}$ was unable to increase the expression of the *CCA1*-promoter-driven Luciferase (Supplementary Fig. 13a). In response to blue light, the activation activity of TCP22, PPK1, and LWD1 was significantly enhanced by CRY2 (Fig. 5j), in contrast to the reconstituted system lacking wild type CRY2, LWD1 (Fig. 5j and Supplementary Fig. 10b) or the reconstituted system expressing mTCP22$^{12STA}$ instead of wild type TCP22 (Supplementary Fig. 13b, c). LWD2 exhibited similar activity to LWD1 (Supplementary Fig. 14a–c). Notably, without TCP22, LWD1 was not sufficient to activate *CCA1* (Fig. 5i, final column Supplementary Fig. 10c), suggesting that, similar to LWD1, TCP22 is also necessary for CRY2 dependent induction of *CCA1* transcription.

## Discussion

Since cryptochromes were found to regulate the circadian clock[9,10], the mechanism linking light perception by crypto-chromes to the core circadian oscillator has not been fully understood. In this study, we show that CRY2 can form photo-bodies with TCP22 in response to blue light. CRY2 and TCP22 promote the activation of *CCA1* suggesting a possible mechanism by which blue light signals could influence the circadian oscillator via blue-light dependent photobody formation rather than blue-light dependent protein–protein interactions[15–17]. Genetic evidence and reconstitution experiments in vitro or in HEK cells suggest that PPKs and the *CCA1* promoter may influence pho-tobody formation and function and that LWDs may be required for transcriptional activity. Nevertheless, further work would be needed to test whether these are components of photobodies in plant cells in physiologically relevant conditions and to determine whether photobody formation is required for *CCA1* activation (Fig. 6a).

Our study further demonstrated that, in addition to being a photoreceptor, the plant CRY2 protein like animal crypto-chromes, functions as a regulator of a core oscillator component (*CCA1*) in blue light. CRY2, TCP22, PPK1, and LWD1 appear to constitute a positive arm to activate the core oscillator *CCA1* in blue light (Fig. 6b).

Intriguingly, we assessed the properties of CRY2-reated photobodies utilizing fluorescence protein tags, immunostaining, and in vitro reconstitution assay in this study, and found that CRY2 containing photobodies exhibit many characteristics common to liquid-liquid phase separation condensates, suggest-ing LLPS may regulate light input to the circadian clock. It is tempting to further speculate that other photoreceptors or sig-naling proteins are also involved in LLPS regulation of the cir-cadian clock. For example, PIFs, and even phytochromes may be involved in phase-separation responses to regulate the blue light input to the clock, as they are also photobody proteins interacting with cryptochromes and exhibiting the function of regulating the circadian clock in blue light[37–39].

We observed *CCA1* was arrhythmic in *cry1cry2* background under blue light, which implies that cryptochromes are important circadian photoreceptors, that along with their interacting pro-teins mediate blue light input and calibrate the rhythm of the core oscillator component *CCA1*. This hypothesis is consistent with the recent report that *CCA1* is also arrhythmic in the triple mutant (*lwd1lwd2ttg1*) of CRY2-interacting LWDs[40]. However, as reported previously, in the complicated circadian clock net-work, the core oscillator component *CCA1* is affected by other proteins/pathways, especially during the oscillation maintaining stage followed by light input[41,42]. For example, the output of the circadian clock *CAB::LUC* is still rhythmic but with a lower amplitude in *cry1cry2* mutant[10]. Further studies on how photo-bodies or LLPS regulate the circadian clock are required.

## Methods

**Plant materials and growth conditions**. All *Arabidopsis* lines used in this work are of the Columbia (Col-4) accession. The *ppk124* mutants are described previously[16], *cry2*, *cry1cry2* mutants, *GFP*, *GFP-CRY2*, and *CRY2-GFP* lines are described previously[15,19,27], *tcp20* (SALK_088460C) and *tcp22* (SALK_045755C) single mutants were obtained from ABRC (http://www.arabidopsis.org/index.jsp). The *tcp20tcp22* double mutant was prepared by a genetic cross of *tcp20* and *tcp22*. To generate *Myc-TCP22*/WT lines, we cloned the full coding sequence of TCP22 into pDT1[16] at the XmaI site using the In-Fusion cloning method, resulting in the expression of Myc-TCP22 driven by the *ACTIN2* promoter. To generate *Myc-PPK1* lines, the plasmid *35S::Myc-PPK1* was prepared by cloning the *PPK1* cDNA into pEGAD-Myc vector[20] in the EcoRI and XhoI sites downstream from the 35S promoter. To generate the *proTCP22::Myc-TCP22* lines, the upstream 2000 bp fragments of the translational start sites were cloned by PCR, *Myc-TCP22* was

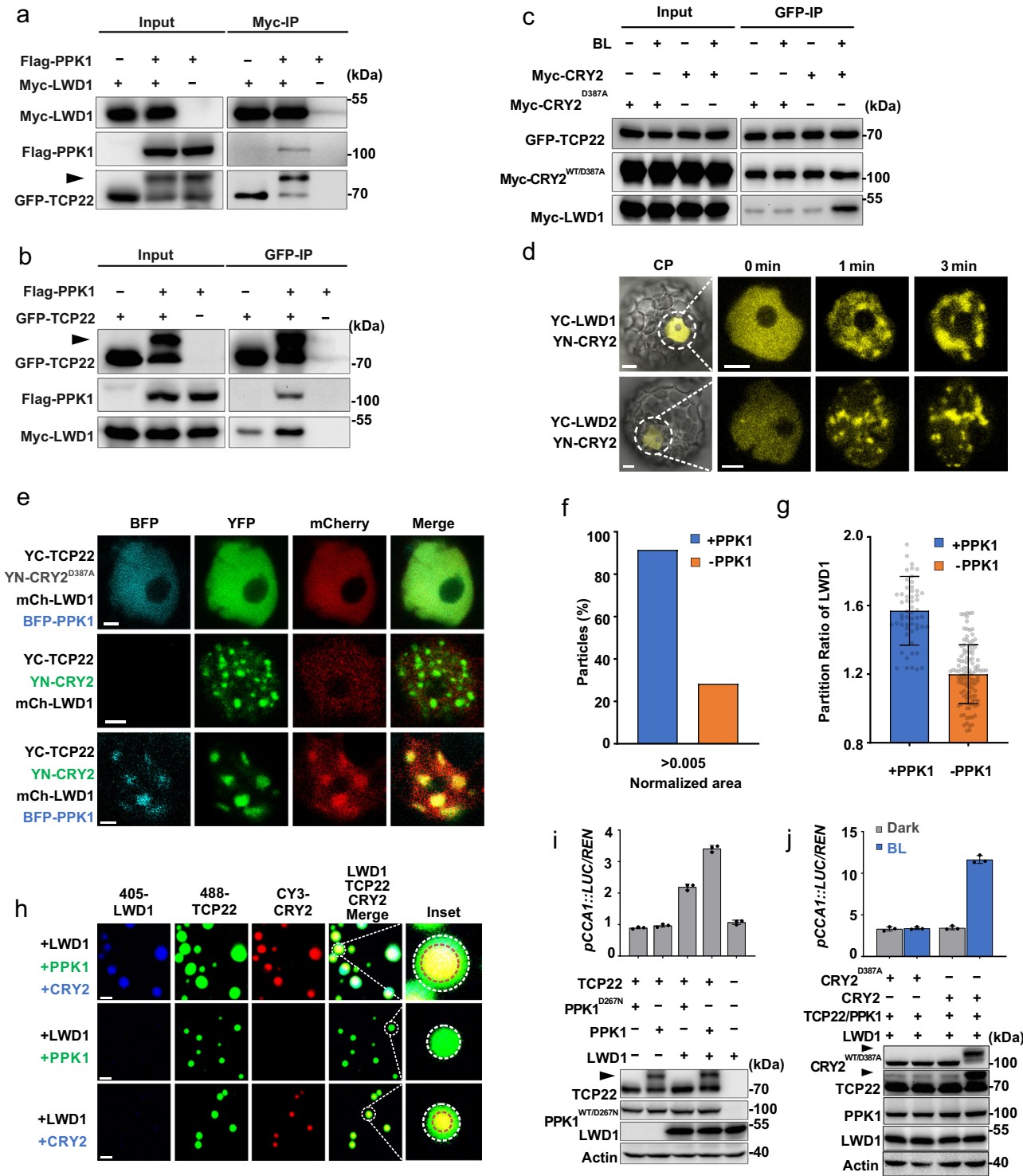

cloned from PCI(neo) *Myc-TCP22*,*TCP22* promoter was bridged with *Myc-TCP22* by overlap PCR and then inserted into pCambia3301 using SacI and BamHI. The Ti plasmid was transformed into the *Arabidopsis* (Col), *tcp20tcp22* or *cry1cry2* using the floral-dip method[43]. The transgenic T1 populations were screened on composite soil irrigated with the 1‰ Basta solution (V/V) . The screened plants were grown in walk-in growth chambers at 22 °C, 65% relative humidity under cool white fluorescent tubes under Long-Day (LD) photoperiods (16 h light of 120 μmol m$^{-2}$ s$^{-1}$/8 h dark).

**HEK-293T cell culture and transfection**. HEK-293T cells (ATCC,ATCC®CRL-11268TM) were cultured in DMEM (Biological Industries, 06-1055-57-1ACS) supplemented with 10% (v/v) FBS (Biological Industries,04-001-1ACS), 100 U/mL penicillin, and 100 mg/mL streptomycin (Hyclone,SV30010) in humidified 5% (v/v)

$CO_2$ in air, at 37 °C.HEK-293T cells were seeded at a density of $3 \times 10^5$ cells per well in a six-well plate and transfected using PEI-max or the Lipofectamine 3000 methods[27]. For the transfection with PEI-max, different combinations of plasmids (2–3 μg/construct) and 6 μL 0.1 mg/mL PEI-max (Polysciences, Inc) aqueous solution were diluted into a total volume of 150 μl of Opti-MEM (Gibco, 11058021) and then kept at room temperature for 5 min before being applied for cells per well. For the transfection using Lipofectamine 3000 (ThermoFisher, L3000015), we performed the transfection according to the manufacturer's instructions.

**Plasmids constructs**. The HEK-293T cell protein expression system was utilized. *Myc*, *Flag* and *GFP* were inserted into pCI (neo) (Promega, E1841) using the EcoRI restriction site. The coding sequences for *TCP22*, *CRY2*, *PPKs*, and *LWDs* were then cloned into pCI (neo) Myc, pCI (neo) Flag or pCI (neo) GFP using XbaI and

**Fig. 5 CRY2, TCP22, LWD1, and PPK1 promote the expression of *CCA1* in a reconstituted system.** Co-immunoprecipitation assay showing the phosphorylated TCP22 interacted more strongly with LWD1 with Myc-LWD1 immunoprecipitation (**a**) or GFP-TCP22 immunoprecipitation (**b**). Two independent experiments showed similar results. **c** Co-immunoprecipitation assay showing that CRY2 enhanced the interaction of TCP22-LWD1 under blue light (30 μmol m$^{-2}$ s$^{-1}$). Two independent experiments showed similar results. **d** Photobody formation of CRY2-LWDs in Col4 protoplasts. Scale bar, 5 μm for the CP (Complete protoplast); 3 μm for the nucleus. Five independent experiments showed similar results. **e** Photobody formation of indicated plasmids co-expressed in Col4 protoplasts. Five independent experiments showed similar results. Scale bar, 2 μm. **f** Size distribution of photobodies in (**e**). ($n = 12$ cells for +PPK1 and $n = 10$ cells for -PPK1). **g** Partition ratio of (**e**). Data are presented as mean ± SD ($n = 12$ cells for +PPK1 and $n = 10$ cells for -PPK1) (**h**) Photobody formation of the purified His-TCP22 (8 μM), His-LWD1 (5 μM) and His-CRY2 (2 μM) labeled with iFlour 488, iFlour 405 and CY3, respectively. After mixing with 8 μM PPK or 8 μM BSA, photobodies of each sample were observed by confocal microscope. Scale bar, 3 μm. Five independent experiments showed similar results. **i** Dual luciferase assay showing the phosphorylated TCP22 was dependent on LWD1 to activate the expression of *CCA1* in HEK-293T cells. Expression levels of protein were estimated by immunoblot. Data are presented as mean ± SD ($n = 3$ independent experiments). **j** Dual luciferase assay showing CRY2 promoted the expression of *CCA1* coordinating with TCP22, PPK1, and LWD1 under blue light (BL, 30 μmol m$^{-2}$ s$^{-1}$). Data are presented as mean ± SD ($n = 3$ independent experiments).

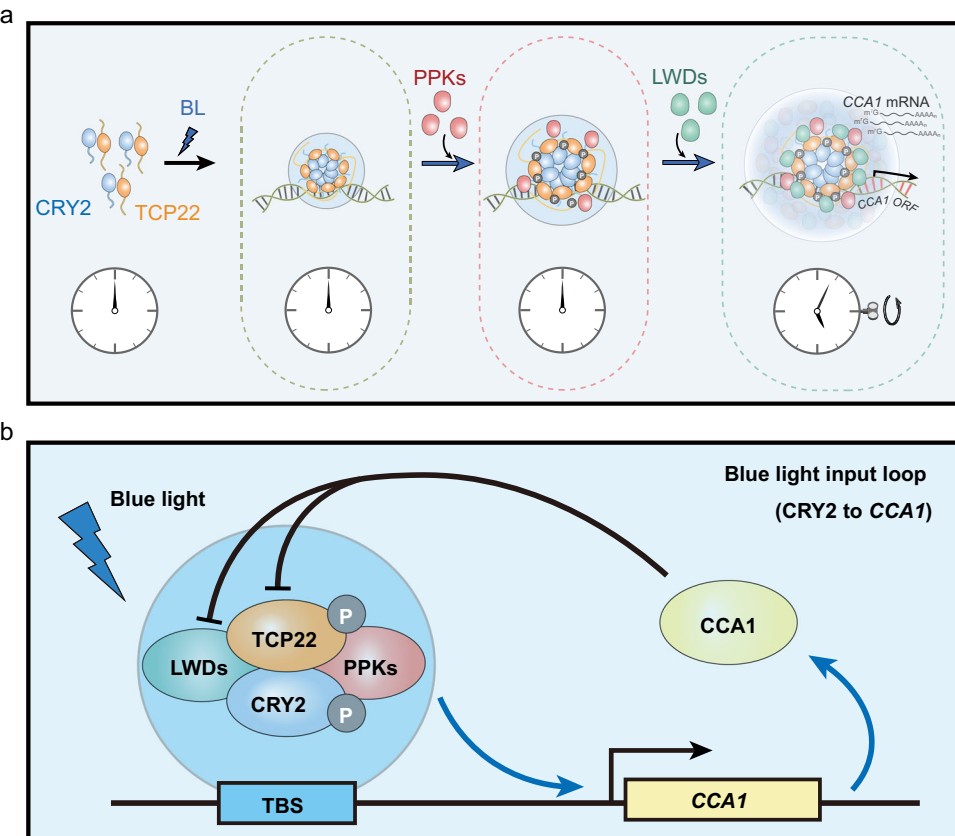

**Fig. 6 Hypothetical model for *CCA1* activation by CRY2. a** Possible mechanism by which CRY2 mediates blue light input to the clock central oscillator via photobody formation. **b** Our study demonstrates that CRY2, TCP22, PPKs, and LWDs could positively regulate *CCA1* under blue light.

XmaI restriction sites. pNL2.2 (Promega, N1071) was modified by replacing the *HygR* gene with *Renilla*[27], and *pCCA1::LUC* was cloned into pNL2.2 and used for the dual luciferase assay.

For the split luciferase complementation assay, the dual promoter vector pDT1[16] was used as a backbone. *nLUC* fragment was bridged with the *TCP22* coding sequence using (GGGS)₃-(NLS)₃ linker by overlapping PCR and then inserted into the pDT1 plasmid using SpeI/XmaI restriction sites. *PPK1* or *LWD1* sequence was bridged with *cLUC* using (GGGS)₃-(NLS)₃ linker by overlapping PCR and then inserted into the pDT1 plasmid using AscI/MfeI restriction sites.

For the in vivo photobody formation assay, pXY103-nYFP and pXY105-cYFP[44] were used as backbones. Full length coding sequences of *TCP22*, *CRY2*, *LWD1*, and *PPKs* were ligated into pXY105-cYFP or pXY103-nYFP using BamHI and XbaI restriction sites respectively. *BFP* (Beyotime, D2701) or *mCherry*[27] was bridged with the indicated cDNA (*PPK1* or *LWD1*). The fused fragments were integrated into pXY105 using KpnI and XbaI restriction sites to generate BFP-PPK1, mCherry-PPK1, mCherry-LWD1. To construct plasmids containing *ProTCP22:: cYFP-TCP22* or *ProCRY2:: nYFP-CRY2*. The upstream 2000bp genomic fragments from translational start site of TCP22[21] or 1516 bp genomic fragments from translational start site of CRY2 were amplified as promoters. PCR products of *cYFP-TCP22* or *nYFP-CRY2* from above pXY plasmids were bridged with their

promoters by overlapping PCR and then insert into the pCambia3301 vector using SacI and BamHI restriction sites. Primers used for vector construction are listed in Supplementary Table 1.

For the in vitro photobody formation assay, a His tag was fused to the N-terminals of the *CRY2*, *PPK1* and *LWD1* coding sequences by PCR. The fragments were then inserted into pCI (neo) using EcoRI and XmaI restriction sites respectively to generate pCI (neo) His-CRY2, pCI (neo) His-PPK1 and pCI (neo) His-LWD1. The *TCP22* coding sequence was inserted into pET-N-His-TEV (Beyotime, D2905) using the EcoRI and BamHI restriction sites.

The *mTCP22*[12STA] fragment was generated by overlapping PCR to introduce mutations at the protein level that will not be phosphorylated by PPK1. The fragment was then inserted into plasmids using the same cloning strategy as *TCP22*.

**Phosphorylation and dephosphorylation assay in HEK-293T cells.** Cells co-expressing pCI (neo) GFP-TCP22 and pCI (neo) Myc-PPKs were lysed 36 h post transfection. Lysates were treated with or without Lambda Protein Phosphatase (NEB P0753S) in a reaction conditions [50 mM HEPES, 100 mM NaCl, 2 mM DTT, 0.01% Brij35, 1 mM MnCl₂][45]. After incubated at 30 °C for 1 h, the samples

were boiled at 100 °C for 10 min and analyzed by immunoblots probed with anti-GFP (1: 3000, MBL, 598) or anti-Myc (1: 3000, MBL, M047-3) respectively.

**Co-immunoprecipitation (Co-IP) assays**. In Co-IP experiments using HEK-293T cells, after exposed by blue light or kept in dark for 3 h, the transfected cells were washed with PBS pH7.4 (Gibco, 8118044), digested with TrypLE™ Express (1×) (Gibco,12605-028) at 37 °C for 5 min. The cells were then centrifuged at 800 g for 5 min and the supernatant discarded. The cell pellets were lysed with Pierce IP Lysis Buffer (87787, Pierce) with 1× EDTA-free Protease Inhibitor Cocktail Tablets (4693159001, Roche) and 1×PhosSTOP inhibitor cocktail (Roche) and incubated on ice for 15 min. The mixtures were then centrifuged at $14000 \times g$ for 10 min at 4 °C to remove cell debris. The supernatant was mixed with 20 μL GFP trap beads or Myc agarose beads, incubated with vertical blending at 4 °C for 2 h. The beads-protein complex was washed 4 times with washing buffer [20 mM HEPES (pH 7.5), 40 mM KCl, 1 mM EDTA] and denatured by mixing thoroughly with 30 μL 4× Loading buffer and heating at 100 °C for 10 min[16].

**Yeast two-hybrid assays**. For analysis of the interaction of TCP22 and CRY2 or PPKs, the prey vector pGADT7 (Clontech, 630442) expressing TCP22 fused to the GAL4 activation domain and the Bait vector pBridge (Clonetech, 630404) expressing CRY2 or PPKs fused to the GAL4 DNA binding domain were used, AD and BD fusion protein vectors were co-transformed into the yeast strain Y190.Colonies were selected on plates (SD-LW). After culturing in SD-LW medium for 12 h (Dark, overnight), transformants were sub-cultured into fresh YPDA medium and kept in dark or irradiated with blue light (BL, 30 μmol $m^{-2}$ $s^{-1}$) for 3 h. β-galactosidase activity was measured using chlorophenol red-b-D-galactopyranoside as a substrate and Miller Units were calculated according to the Clontech Yeast Protocols Handbook.

**Dual luciferase assay in HEK-293T**. The *CCA1* promoter (- 635 to - 1 relative to the translational start site) was inserted into the modified PNL2.2 vector to drive the expression of luciferase. HEK-293T cells were co-transfected into the desired vectors using the Lipofectamine 3000 method. Samples were kept in the dark or exposed to blue light (30 μmol $m^{-2}$ $s^{-1}$) for 3 h before being lysed, the experiment was performed according to the user manual for (Promega, E1910) kit.

**In vivo Phosphorylation assay**. Seedlings of different genotypes (*Myc-TCP22*/ WT, *Myc-TCP22/cry1cry2*) were cultured in the dark on 1/2 MS solid medium for 6 days. These etiolated seedlings were then exposed to blue light (45 μmol $m^{-2}$ $s^{-1}$) for the indicated time before samples were collected. For the reversible blue-light induced phosphorylation assay, the seedlings were collected at the specified time points during 24 h of exposure to blue light. The remaining plant materials were then transferred to dark conditions and samples were collected at the indicated time. The collected samples were ground in liquid nitrogen and mixed with 4× sample buffer. The mixture was then vortexed and boiled at 100 °C for 10 min to extract the protein. The protein was detected on a 10% SDS–PAGE and transferred to Immobilon NC Transfer Membrane (HATF00010, Millipore) to perform immunoblotting.

TCP22 phosphorylation in the presence (WT) or absence of PPKs (*ppk124*) in response to the blue light (30 μmol $m^{-2}$ $s^{-1}$) was performed using the AGROBEST method[46]. Four-day-old *Arabidopsis* including WT, and *ppk124* seedlings were infected with Agrobacterium C58C1 (pTiB6S3ΔT)[H] carrying the vector pDT1 *ACTIN2::4×Myc-TCP22 UBQ10::LUC* for 3 days in a 16 h/8 h light/dark cycle (75 μmol $m^{-2}$ $s^{-1}$). The transfected seedlings were transferred to 1/2 MS liquid medium in the presence of 100 μM Timentin overnight in the dark environment. Seedlings growing in blue light (30 μmol $m^{-2}$ $s^{-1}$) were harvested at the indicated time point for use in immunoblotting.

**Bioluminescence assay and data analysis**. The fragment of *CCA1* promoter (−984 to −1 relative to the translational start site) used in this assay was described previously[21]. The promoter fragment was cloned and fused with the *LUC* gene by overlapping PCR and inserted into vector pCAMBIA-3301 to construct the *pCCA1::LUC* by SacI and BamHI. The vector was transformed into WT, *cry1cry2* or the *ppk124* mutant. *pCCA1::LUC*/WT was then crossed with *CRY2-GFP*/WT, *Myc-PPK1*/WT, *Myc-TCP22*/WT to generate *pCCA1::LUC/CRY2-GFP*, *pCCA1::LUC/Myc-PPK1*, *pCCA1::LUC/Myc-TCP22*. To construct *pCCA1::LUC/Myc-TCP22/ppk124* and *pCCA1::LUC/Myc-TCP22/cry1cry2*, pDT1 *Myc-TCP22*[16] (hygromycin) was transformed into *pCCA1::LUC/ppk124* and *pCCA1::LUC/cry1cry2* using the floral-dip method. For detecting the luciferase activity during the processes of transferring from dark to blue light, several seedlings ($n \geq 30$) of different genotypes were grown on 1/2 MS solid medium containing 3 mM D-Luciferin in a well of a 96-well plate in the dark for 5 d, then transferred to blue light (10 μmol $m^{-2}$ $s^{-1}$) and with luciferase signals being collected at 20 min intervals. Data were normalized using values of the first time point.

The circadian clock was tested under continuous blue light or red light. Seedlings of different phenotypes were grown on 1/2 MS solid medium containing 3 mM D-Luciferin in 96-well plates in 12 h dark/12 h light (75 μmol $m^{-2}$ $s^{-1}$ white light) treatment for 3 days. These seedlings then transferred into continuous red

light (5 μmol $m^{-2}$ $s^{-1}$) or continuous blue light (5 μmol $m^{-2}$ $s^{-1}$) for 4 days, subsequently, LUC activity was detected every 1 h for the indicated time under continuous red or blue light. Data for LUC activity was collected and analyzed using FFT-NLLS, which is available at https://biodare2.ed.ac.uk. Bioluminescent intensity at each time point was normalized with the intensity of first time point, and then normalized to the maximal value[47].

For blue light pulse assay, seedlings of different phenotypes were grown on 1/2 MS solid medium containing 3 mM D-Luciferin in 96-well plates under 12 h dark/ 12 h light (75 μmol $m^{-2}$ $s^{-1}$ white light) treatment for 3 days and then transferred into continuous red light (10 μmol $m^{-2}$ $s^{-1}$) for 4 d, then luciferase activity was detected at 20 min intervals using the Centro XS3 LB 960 Luminometer under the same red light treatment conditions. Blue light (15 μmol $m^{-2}$ $s^{-1}$) was pulsed from 144 h to 168 h. LUC activity data was collected and analyzed using FFT-NLLS.

To detect the activation of *CCA1* by mTCP22[12STA], which can't be phosphorylated by PPK1. Bioluminescence analysis of *pCCA1::LUC* expression in indicated genotypes using the Agrobest method[46]. Four-day-old seedlings containing *pCCA1::LUC* were infected with A. tumefaciens strain C58C1(pTiB6S3ΔT)H carrying *Myc-TCP22*, *Myc* or *Myc-mTCP22*[12STA] in ABM-MS co-cultivation medium, respectively. At 3 dpi(days post infection), seedlings were transferred to MS medium containing 100 μM Timentin and 0.5 mM luciferin in a 96-well plate. Seedlings were irritated by blue light (10 μmol $m^{-2}$ $s^{-1}$). Luciferase signals were collected at 20 min intervals.

**Sample preparation for mass spectrometry (MS) analyses**. For the analysis of the interaction of CRY2 and TCP22 in *Arabidopsis* by mass spectrometry, the GFP-CRY2 fusion protein was purified from seedlings overexpressing GFP-CRY2 using the GFP-trap method[48]. *Arabidopsis* seedlings overexpressing GFP-CRY2 were grown on 1/2 MS solid media under LD conditions (16 h light/ 8 h dark) for 10 days. After 2 days dark treatment, seedlings were collected after exposure to 30 μmol $m^{-2}$ $s^{-1}$ blue light for 48 h. Tissues were ground in liquid nitrogen, homogenized in NEB buffer [20 mM HEPES (pH 7.5), 40 mM KCl, 1 mM EDTA, 1% Triton X-100, 1 mM PMSF] and centrifuged at $16,000 \times g$ for 15 min. The supernatant was mixed with GFP-trap beads to co-precipitate the proteins inter-acting with GFP-CRY2. The precipitated proteins were then eluted using elution buffer [0.1 M glycine (pH 2.5)], neutralized with neutralization buffer [1 M This-HCl (pH 10.5)], and separated using SDS-PAGE gel for digestion.

For analysis of the phosphosites of TCP22 phosphorylated by PPK1 or PPK1[D267N] in HEK-293T cells, GFP-TCP22 was precipitated using a similar method to the Co-IP assay in HEK-293T cells, and samples prepared for MS analysis using the same procedure used for proteins interacting with GFP-CRY2.

**In-gel digestion**. All samples were separated using SDS-PAGE gels and stained with Coomassie brilliant blue (B7920-10G, Sigma-Aldrich). Bands of interest were extracted from the gel using a scalpel. After dehydration with acetonitrile (75-05-8, Optima), proteins in the gel slices removed were reduced with 10 mM DL-Dithiothreitol (3483-12-3, Sigma-Aldrich). 55 mM iodoacetamide (144-48-9, Sigma) was added to alkylate the reduced sulfydryl groups. After washing with 50 mM ammonium bicarbonate (40867-50G-F, Sigma) and dehydrating with acetonitrile, the proteins were digested with trypsin (V5111, Promega) and chy-motrypsin (V106A, Promega).

**nano-UPLC-MS/MS analysis**. For analysis of the phosphosites of TCP22 phos-phorylated by PPK1 or PPK1[D267N], the peptides were resuspended with solvent A water (7732-18-5, Optima) containing 0.1% formic acid (64-18-6, Sigma-Aldrich)) and analyzed by on-line nanospray LC-MS/MS on an Orbitrap Fusion Lumos coupled with an ACQUITY UPLC M-class System (Waters, Eschborn, Germany). For one LC-MS/MS run, 1ug peptide sample was loaded onto the trap column (nanoEaseTM M/Z Symmetry C18,180 μm × 2 cm, waters) for 10 min with a flow of 3 μL/min at 3%B [B: ACN (75-05-8, Optima)with 0.1% formic acid] and separated on the analytical column (nanoEaseTM M/Z HSS C18, 75 μm × 25 cm, waters) in a linear gradient of 3% B to 35% B over 78 min. The column flow rate was maintained at 500 nL/min. The electrospray voltage of 2.2 kV versus the inlet of the mass spectrometer was used.

For DDA, the mass spectrometer was automatically switched under MS and MS/MS mode in 3.5 s cycles. MS1 mass resolution was set at 60000 with m/z 355–1550. The dynamic exclusion was set as $n = 1$, and the dynamic exclusion time was 30 s. AGC target is 4e5 and max injection time is 50 ms. MS2 resolution was set as 30,000 under HCD mode. The AGC target is 5e4, max injection time is 100 ms. DDA raw data were analyzed using the Thermo Proteome Discoverer (2.2.0.388).

For PRM, the resolution of MS1 full scan was set to 60,000. The AGC target was set to 4e5 and the maximum injection time was set to 50 ms. The resolution of multiple PRM scans (MS2) was 30,000. The AGC target was set to 5e4 and the maximum injection time was set to 300 ms. Precursors of each phosphorylated peptides were selected by the quadrupole mass analyzer (1.2 Da isolation window). Raw PRM data were analyzed using Skyline daily (version 3.5)[27] to extract and calculate the transition peak areas. The parameters were as follows: the mass difference within ±20 ppm and dot-product (dotp) score >0.7.

For analyzing the interaction of CRY2 and TCP22 in *Arabidopsis* by Mass Spectrometry. Raw files were acquired using a method similar to the above. Raw files

were analyzed together using Maxquant (1.6.2.10)[27]. CRY2-interacting Proteins were identified using a target-decoy approach by searching all MS/MS spectra against a concatenated forward/reversed version of TAIR10_pep_20101214 sequence database. The parameters were as follows: strict trypsin specificity, allowing up to two missed cleavages, minimum peptide length was seven amino acids, carbamidomethylation of cysteine was a fixed modification, N-acetylation of proteins and oxidation of methionine were set as variable modifications. Peptide spectral matches and protein identifications were filtered using a target-decoy approach at a false discovery rate of 1%. 'Match between runs' was enabled with a match time window of 0.7 min and an alignment time window of 20 min[49]. Relative, label-free quantification (LFQ) of proteins was integrated into MaxQuant. Normalized intensity was generated according to the algorithms described in Cox et al.[49]

**Protoplast isolation and purification.** Isolation and purification methods are based on "Tape-Arabidopsis-Sandwich"[50] with some modifications. Leaves of 3-week-old plants in LD (16 h light, 8 h dark) were cut and peeled away from the lower epidermal surface using breathable tape (3 M Micropore™, 1530C-0) and colorful tape (VBWINTAPE). The remaining leaves were transferred to the enzyme solution [20 mM MES pH 5.7, 1.5% (w/v) cellulase R10, 0.4% (w/v) macerozyme R10, 0.4 M mannitol, 20 mM KCl and 10 mM CaCl$_2$]. After 2 h digestion at room temperature, the solution was filtered through a 75-micron nylon mesh (Calbiotecm® Cat: 475855-1 R) and centrifugated at $100 \times g$, 4 °C for 2 min to pellet the protoplasts. Then the protoplasts were washed with W5 buffer [2 mM MES pH 5.7, 154 mM NaCl, 125 mM CaCl$_2$, 5 mM KCl, 5 mM D-glucose] twice and incubated on ice for 30 min. For PEG-Ca$^{2+}$ protoplast transformation, protoplasts were washed with ice-cold MMG buffer [4 mM MES pH 5.7, 0.4 M mannitol, 15 mM MgCl$_2$] twice and resuspended with 100 µL MMG buffer per sample. The indicate volumes of plasmids were added and mixed with protoplasts. (For BiFC assay, 7 µg of each plasmid was added; for untagged CRY2 inducing assay, 3 µg cYFP-TCP22 and TCP22-nYFP with 10 µg CRY2 were added; for CRY2-GFP photobody formation assay, 15 µg plasmid were added; for experiment containing mCherry-PPK1, only 1 µg plasmid was added.) The mixture was then mixed with 110 µL PEG-Ca$^{2+}$ buffer [40% (w/v) PEG4000, 0.2 M mannitol, 100 mM CaCl$_2$] and kept at room temperature for 5 min, washed twice with W5 buffer at room temperature. The protoplasts were finally resuspended with 1 mL W5 buffer and transferred into a six-well plate and incubated for at least 12 h in dark for next experiments.

**Confocal microscope imaging.** For TCP22-PPKs BiFC assay, the fluorescence images were captured using a LSM880, Zeiss confocal microscope. Image analyses were performed using the Zen software (Zeiss) and processed with Adobe Photoshop. For the other assays, all pictures were captured using a TCS SP8X, Leica, 63× oil objective, PMT/HYD detector. Image analyses were performed using the Leica software (LAS X) and processed with Adobe Photoshop.

**Nuclear extraction and immunostaining.** 10-d-old red light-grown seedlings were incubated in 50 µM MG132 for 12 h and exposed to 30 µmol m$^{-2}$ s$^{-1}$ blue light for indicated time before nuclear extraction. Nuclear extraction was performed as Yu described with some modifications[51], briefly, seedlings were collected and fixed in 4% formaldehyde for 20 min after blue light treatment, washed twice with PBS, chopped using a razor blade, and suspended in sorting buffer [100 mM Tris-HCl, pH 7.5, 50 mM KCl, 2 mM MgCl$_2$, 0.05% Tween 20, and 5% sucrose], and resuspended with buffer 2[125 mM sucrose,10 mM Tris-HCl, pH 9.5, 10 mM KCl, 0.1%Triton-100], added onto buffer 3[850 mM sucrose,10 mM Tris-HCl, pH 9.5, 10 mM KCl, 0.1%Triton-100) and centrifuged by 15000 rpm at 4 °C for 30 min to precipitate the nuclei. Nuclei were incubated with anti-CRY2 antibody (1:100, prepared in our lab[52]) and anti-Myc (1:100, MBL, M192-3) overnight, washed with 0.2%PBST 3 times, incubated with CY3 conjugated goat anti-rabbit IgG (1:500, Beyotime, A0516) and Alexa Fluor 488 conjugated goat anti-mouse IgG (1:500, Beyotime, A0428). Images were captured at 63× oil magnification using a Leica confocal microscope (TCS SP8X, Leica). Immunoblots were used to quantify expression levels of CRY2 and Myc-TCP22. CRY2 and Myc-TCP22 were detected with anti-CRY2 and anti-Myc, respectively.

**Split luciferase complementation assay.** The method of *Arabidopsis* protoplast transient expression is the same as used for the BiFC assay mentioned below. Protoplasts from Col4, *cry2* mutant were transfected with pDT1 plasmids[16] dual trans-gene vectors containing nLUC-TCP22 and cLUC-PPK1. The transfected protoplasts were incubated in 96-well plates in the dark at 22 °C overnight. Luciferin was mixed with protoplasts to a final concentration of 1 mM in each well of a 96-well plate. After incubated for 40 min at room temperature, the luminescence was detected under blue light (10 µmol m$^{-2}$ s$^{-1}$) for the indicated time using a Centro XS3 LB 960 Luminometer at a 10 min intervals[53].

**In vivo photobody formation assay.** Protoplasts transfected with the indicated genes were transferred to 20 mm glass bottom dishes (Glass Bottom Cell Culture Dish, NEST, Cat.NO. 80100). The laser of the confocal microscope (TCS SP8X, Leica) was applied to activate light induced photobody formation. Though CRY2 has a better absorption ability under 488 nm laser[27], the extremely rapid speed for the CRY2-TCP22 complex to form condensate under the same light intensity

renders it inappropriate for confocal imaging. We noted that CRY2-TCP22 condensates would form in less than 0.8 s under 488 nm laser. Under such circumstance, the 514 nm lasers were used to activate the formation of CRY2-TCP22 condensates, where the CRY2 has relative low absorption and YFP has the highest absorption (See Fluorescence Spectra Viewer (https://www.thermofisher.com/cn/zh/home/life-science/cell-analysis/labeling-chemistry/fluorescence-spectraviewer.html).

The argon 514 nm laser at 2% was used in most of the assays in this paper and also for untagged CRY2 and CRY2-GFP activation. Indicated on/off time intervals described in corresponding figure legend or continuous light, were used to activate the photobody formation. Strictly speaking, '0 s' was rationally defined as an image that was captured under absolute dark but confocal microscope needs 0.8 s to finish each shot. We therefore set the time point that the first image was captured under blue–green or blue laser as '0 s'.

**Fluorescence recovery after photobleaching (FRAP).** After activation of indicated time, condensates were bleached with 100% power using 488 nm or 514 nm laser and the recovery images were captured under 2% power of the indicated lasers. The florescence intensity of bleached condensates was calculated using the LAS X software. Three parameters were collected in each FRAP assay, which were the intensity of ROI (region of interest), background and reference (non-bleached region). Intensity of ROI and reference were collected at the indicated time point and the intensity of background was collected in the pre-bleach time. The data were normalized with the formula:

$$I_{c1}(t) = I(t) - B$$

$$R_{c1}(t) = R(t) - B$$

Where the $I$ and $R$ are the intensity of region of Interest and Reference, separately. $B$ is the intensity of the background and $t$ is the specific time point. Further, the $I_{c1}$ and $R_{c1}$ were normalized with the formula:

$$I_{c2}(t) = I_{c1}(t)/R_{c1}(t)$$

Finally, to set the intensity of pre-bleach as 1, the formula was used as below:

$$\text{NormalizedIntensity} = I_{c3}(t) = I_{c2}(t)/I_{c2}(p)$$

Where the $p$ means the time point of pre-bleach.

**Reversible light-induced photobody formation.** For the light cycle assay, the TCS SP8X, Leica confocal microscope was set in a dark room, a continuous ~16 µW 514 nm single laser was used to activate the photobody formation of protoplasts for 10 min. The laser was then turned off and the protoplasts were left the dark environment for 10 min to allow the condensate to dissolve. For the protoplasts that were co-expressing mCherry-PPK1, dual lasers with wavelengths of 514 and 581 nm were used to capture images at the indicated time points given in Fig. 3f. Similarly, 10 min ~16 µW single 514 nm laser activation and 10 min dark recovery were undergone to insure the absolute consistency of activation conditions.

For the activation and recovery of condensates of CRY2-GFP, continuous 488 nm laser was used to activate the CRY2-GFP. 488 nm laser with ~0.8 s capture time was used to capture the recovery process in the indicated time point.

**1,6-hexanediol treatment.** For the in vivo 1,6-hexanediol inhibition assay, 1,6-hexanediol powder was dissolved in W5 buffer to a concentration of 10% (w/v). Before microscope capturing, 10% 1,6-hexanediol or W5 buffer (see Protoplast isolation and purification) was added in the center of the glass dish in advance. An equal volume of W5 buffer containing protoplasts were added and mixed immediately by gently pipetting with a 100 µL tip. The mixture was incubated in the dark for 5 min and images were then captured using the confocal microscope.

For the in vitro 1,6-hexanediol inhibition assay,1,6-hexanediol powder was dissolved in observe buffer (see in vitro photobody formation assay) to s concentration of 40%(w/v). This solution was added to the purified protein solution to a final concentration of 10% (w/v). After a 5 min incubation period the mixture was observed by confocal microscopy.

For 1,6-hexanediol treatment of immunostaining assay, seedlings were incubated in 10% (w/v)1,6-hexanediol dissolved by 1/2 MS under red light (30 µmol m$^{-2}$ s$^{-1}$) for 30 min, then the seedlings were transferred into blue light (30 µmol m$^{-2}$ s$^{-1}$) for 30 min, nuclear extraction and immunostaining were conducted as described above.

**Protein expression and purification.** The HEK-293T cells transfected with pCI (neo) His-CRY2 were irradiated with blue light (100 µmol m$^{-2}$ s$^{-1}$) for 3 h or kept in continuous dark before being lysed on ice for 20 min using following buffer: 50 mM NaH$_2$PO$_4$ pH 8.0, 500 mM NaCl, 1% NP40 with EDTA-free protease inhibitor cocktail. The cells were treated under blue light or dark during lysing to maintain the active or inactive status of CRY2. The cell lysate was sonicated on ice with settings 3 s ON/9 s OFF at 30% intensity for 4 min, and centrifuged at 12,000 rpm for 40 min. The soluble protein was purified using Ni-NTA-Sefinose Column and washed with the following buffer: 50 mM NaH$_2$PO$_4$ pH 8.0, 500 mM NaCl, 20 mM imidazole. Column binding protein was eluted in the following

buffer: 50 mM NaH$_2$PO$_4$ pH 8.0, 500 mM NaCl, 500 mM imidazole. Removing imidazole and buffer exchange was performed using Amicon Ultra-15 Centrifugal Filter Unit (Millipore, UFC900308, UFC500308). pCI (neo) His-PPK1 and pCI (neo) His-LWD1 were purified in the same manner as the purification of His-CRY2 in the dark.

cDNA encoding *TCP22* was cloned into pET-N-His-TEV. Recombinant plasmids were transformed into BL21(DE3) *Escherichia coli*. A fresh bacterial colony was inoculated in LB media containing kanamycin and grown overnight at 37 °C and then diluted 1:50 in 500 ml LB and grown to OD$_{600}$ = 0.6–0.8, then induced with 0.2 mM isopropyl β-D-1-thiogalactopyranoside (IPTG) for 20 h at 18 °C. Cells were collected, pelleted, and then resuspended in the following buffer: 50 mM NaH$_2$PO$_4$ pH 8.0, 500 mM NaCl, 10 mM imidazole and supplemented with EDTA-free protease inhibitor cocktail (05892791001, Roche) according to the manufacturer's instructions. The cells were lysed by sonication and then centrifuged at 12000 rpm for 40 min. The soluble protein was purified using Ni-NTA-Sefinose Column (Clontech) and eluted in the following buffer: 50 mM NaH$_2$PO$_4$ pH 8.0, 500 mM NaCl, 500 mM imidazole.

**Protein labelling**. The His-CRY2 protein was fluorescently labelled using His Lite™ Cy3 Bis NTA-Ni Complex (AAT Bioquest®, 12610). The protein and dye were mixed at a 1:1 molar ratio and incubated at 4 °C for at least 2 h. The TCP22 protein was labelled using iFluor™ 488 succinimidyl ester (AAT Bioquest®, Cat:1023). The His-PPK1 and His-LWD1 proteins were labelled using iFluor™ 405 succinimidyl ester (AAT Bioquest®, Cat:1021). The proteins were labelled using the protocol for indicated dye. In brief, the protein was diluted at 1 mg ml$^{-1}$ in PBS and mixed with 100 mM sodium bicarbonate. The reaction was incubated for 15 min at room temperature and then incubated for 1 h on ice. Fluorescently labelled proteins were purified from the unreacted dye substrate by Amicon Ultra-0.5 Centrifugal Filter Unit (Millipore, UFC500308).

**Chromatin immunoprecipitation (ChIP)**. The ChIP assay was conducted using the ChIP kit (EpiQuik™ Plant ChIP Kit, EPIGENTEK) and the corresponding protocol. 3-week-old *Arabidopsis* seedlings grown under long day (16 h light/8 h dark) were collected and wash thoroughly. The samples were cross-linked with 1% formaldehyde. 2 M glycine was added into the cross-link buffer to stop the reaction. Plant samples were then removed from the buffer and frozen by the liquid nitrogen to homogenize and release the nucleus. The remaining steps such as nucleus isolation, protein immunoprecipitation, DNA elusion, and DNA purification were performed according to the protocols provided with the kits. The purified DNA and the DNA before immunoprecipitation (input) were used to conduct qPCR tests. For qPCR tests, SYBR Premix Ex Taq™ II (Tli RNaseH Plus) (TaKaRa) was used for qPCR reaction, using the Mx3000P™ Real-Time PCR System (Stratagene).The primers used in qPCR test has been reported previously[21]. The final data was analyzed by the following step[54]:

$$\text{Adjusted input} = \text{Raw } Ct(\text{input}) - \log_2(\text{fraction of total input})$$

In this assay, the aliquot for qPCR was 1.25% of the total input, and the Adjusted input should be:

$$\text{Adjusted input} = \text{Raw } Ct(\text{input}) - \log_2(1.25) = \text{Raw } Ct(\text{input}) - 0.32$$

And the IP/Input was calculated by:

$$\text{IP/Input} = 2^{Ct(\text{AdjustedInput}) - Ct(\text{sample})}$$

**In vitro photobody formation assay**. In vitro photobody formation assay was performed in observation buffer: 50 mM NaH$_2$PO$_4$ pH 8.0, 500 mM NaCl. Droplets were assembled in glass-bottom cell culture dish (NEST,801001) passivated by overnight incubation with 5% Pluronic F127 (Sigma-Aldrich). Co-condensate of indicated proteins was performed by adding label to the final concentration. BSA (Albumin from Bovine Serum) was used as negative control. The droplet size and numbers were analyzed with Image J.

**DNA-Protein droplets formation**. 30 bp reverse complemented single-stranded DNA primer of *CCA1* promoter containing the TBS or mutant TBS[21] sequence were synthesized and dissolved in the observation buffer to a final concentration of 100 μM. The forward and reverse primers were mixed with the ratio (w/w) of 1:1 and heated at 98 °C for 5 min and transferred to room temperature to anneal for 15 min. For random DNA assay, doble-stranded random DNA containing a random sequence of 8 bases in the position of TBS sequence was synthesized according described procedures[55,56]. The products were purified and concentrated to 100 μM. The double-stranded DNA was labeled with DAPI for 1 h. The DNA solution was then mixed with protein solution to the final concentration required and incubated on ice for 10 min. The mixture was observed in the same condition described in the in vitro photobody formation assay.

**Data analysis of photobody formation assay**. Circularity calculation was based on the method described previously[57]. The images of indicated time point were selected and exported to Image J to analyze the perimeter and the area of photobody. The tiny and indistinct photobodies were not counted. The final circularity

was calculated by the formula:

$$\text{Circularity} = 4\pi \frac{\text{photobody area}}{(\text{photobodyperimeter})^2}$$

The photobodies were more circular if the circularity was closer to 1.

The partition ratio was calculated based on the previous research[57]. The images of indicated time point were collected and analyzed by Image J. The intensity of photobodies and the intensity of background were calculated and the final ratio was fit to the formula:

$$\text{Partition ratio} = \frac{\text{photobody intensity}}{\text{background intensity}}$$

The partition ratio of each time point in BiFC assay was calculated by the formula:

$$\text{Partition ratio}(t) = \frac{\sum_{i=0}^{n} P(i)}{n \cdot N}$$

Where the $P(i)$ represents the intensity of each photobody; $n$ represents the photobody number of each nucleus and $N$ represents the mean intensity of nucleus.

For the in vitro photobody formation assay, the background intensity was defined as the mean intensity of the full image; for the in vivo BiFC assay, the background intensity is represented to the mean intensity of whole nucleus. The partition ratio of samples that failed to form the particles was set to 1.

To count the number of photobodies and calculate the area, the images were analyzed using image J. The photobody area was normalized by the area of the whole image for in vitro photobody formation assay. The area of nucleus was used to normalized the area for in vivo BiFC assay analysis. The threshold number of photobodies for each graph is displayed in the corresponding figure legend.

For size distribution of photobodies, the particles (%) was calculated by [number of photobodies (area >0.75)]/[number of total photobodies].

**Quantification and statistical analysis**. All data were analyzed using GraphPad Prism (version 8.0.2), statistical analysis details were presented in figure legend and Methods. Exact *p*-values of statistical tests were provided in the Source Data file and the figure legends.

**Reporting summary**. Further information on research design is available in the Nature Research Reporting Summary linked to this article.

## Data availability
The source data for Figs. 1–5, Supplementary Figs. 1–14 are provided with this paper as a Source Data file. Other data and materials of this study are available from the corresponding author upon reasonable request. The mass spectrometry data for CRY2-interacting proteomics and phosphorylation sites in TCP22 were have been deposited to the ProteomeXchange Consortium via the PRIDE[58] partner repository with the dataset identifiers PXD032848 and PXD032849, respectively.

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

## Acknowledgements

This work was supported by grants from the National Natural Science Foundation of China (31371411 to Z.Z. and 31972508 to Z.Y.) We would also like to thank the UCLA-FAFU (Fujian Agriculture and Forestry University) Joint Research Center on Plant Proteomics, Haixia Institute of Science and Technology, and Fujian Provincial Key Laboratory of Haixia to apply the Confocal microscope imaging system and nano-UPLC-MS/MS analysis system.

## Author contributions

Z.Z. designed the overall research, wrote the paper. W.M. carried out Co-IP, Dual Luciferase assay, and circadian clock test. J.Z. carried out ChIP analysis, BiFC assay, and confocal imaging; L.Z. carried out protein purification and photobody formation in vitro. Immunostaining assay was carried out by W.M., J.Z., and L.Z.; Z.Y. carried out plant material construction and part of photobody formation detection and data analysis. L.Y., Y.L., and N.Y. processed mass spectrometry (LC-MS) analysis. T.X., T.L., M.B., and X.D. contributed to yeast two hybrid assay and plasmid construction, plant genetic analysis. G.Z. helped to conduct protoplast isolation and transfection. W.M., J.Z., and L.Z. analyzed data and make the publish figures.

## Competing interests

The authors declare no competing interests.
