## [Peer Review File · Nature Communications]

Arabidopsis cryptochrome 2 mediates blue light to form the photobody with TCP22 and inputs the light signal to the circadian clockREVIEWER COMMENTS

Reviewer #1 (Remarks to the Author):

The current manuscript explores an important question with respect to light inputs into the circadian clock in Arabidopsis, namely what is the role of the cryptochromes in this response at the molecular level; this is doubly important since cryptochromes play important roles in the circadian clocks of animals and other organisms. The authors provide evidence of an intriguing novel mechanism in which a combination of Cry2 and transcription factor TCP22 undergoes selective condensation in blue light, further recruiting additional proteins to interact with the oscillator component CCA to alter circadian rhythms in blue light. A particularly elegant feature of this study is that the results were replicated in a recombinant mammalian expression system. Because this blue-light dependent condensation does not require blue-light dependent interaction, but only small conformational change of constitutive complexes, it represents a novel means whereby photoreceptors may provide light input into targeted processes. The data and controls are convincing and well presented. For all these reasons, the study merits publication and is likely of interest to a wide audience.

However, a point which the authors largely gloss over is that cryptochromes, and certainly cry2, do not play any major role in the plant circadian clock, and in fact have even been reported to play light-independent functions in red light and affect features such as amplitude and period length - which definitely do not fit into the present scheme (see Devlin and Kay, Plant Cell, 2000). Cry2 in particular is unlikely to play any significant role in blue light, as it is rapidly degraded and quite simply disappears. Apart from that, dozens of articles show rhythmicity in plants in the absence of crys. So, the statement of the authors implying cryptochromes are the major receptor implicated in plant circadian clock need to be softened. A better approach would be to claim an impact on the clock, along with many other players, and speculate for instance how recruitment of pifs and phytochromes to the condensed phase could provide an additional mechanism of cry role in the plant clock.

Some minor points - one is puzzled why there is so little data regarding Cry1, especially as it has been shown to function redundantly with cry2 in the clock. The authors should at least point out parallels (in mechanism of phosphorylation and interaction with similar partners) in comparison to cry2 function. Also, there are many typos and spelling errors, even in the abstract where the authors manage to mis-spell cryptochrome. This should be attended to in a revision.

Reviewer #2 (Remarks to the Author):

This paper aims to identify the contributions of TCP22 and CRY2 in mediating the circadian clock under blue-light conditions. This research question is of interest because TCP22 and CRY2 have been previously shown to constitutively interact with each other despite the presence or absence of blue light signals. Through the emergence of the LLPS field, the authors hypothesize that LLPS could be used as a biophysical mechanism that facilitates TCP22-CRY2 blue light responses, without the need for TCP22 and CRY2 to physically disassociate. Using a mix of biochemistry, confocal imaging, and proteomics, the authors impressively show that TCP22 and CRY2 phase separate to form photobodies within protoplast cells, and that these photobodies have the ability to regulate transcription of the circadian clock protein CCA1. Additionally, they find that two other proteins, kinase PPK1 and light regulated LWD1, assemble into photobodies and also enable blue light responses by helping with both phosphorylation and transcriptional efficiency of TCP22-CRY2 photobodies.

There are many nice things about this paper however, more experiments and clarifications must be made before it should be published. Additionally, the authors may want to have someone proof read it for grammatical errors and sentence structure.

Comments:

Lines 52–55: The authors mention that the interaction of nCRY2 and cTCP22 formed fluorescent photobodies in response to blue light (figure 1d). However, they use a 514 nm laser, which illuminates green light. Will the authors please explain why they did not use a 488nm laser for excitation.

Lines 82-87: The authors conclude that CRY2-TCP22 LLPS is blue-light specific. However, they never directly test the formation of CRY2-TCP22 condensates under other light sources. To me, this suggests that blue light is sufficient for LLPS but does not rule out the possibility of other light sources triggering their formation. Please have the authors confirm that this is a blue-light specific response.

Lines 122-128: The author's hypothesis for the role of CRY2's IDR is interesting and should be tested. Since they are doing this work in protoplasts, the authors should test how truncation or modifications to the IDR region influence phase separation.

Lines 173-186: To tie the relationship between CRY2-TCP22 condensates and CCA1 transcription together, the authors look to see if the TBS motif of the CCA1 promoter can trigger LLPS in vitro. However, this experiment is flawed/ missing some vital controls. For instance, for supplemental figure 6, why is there no DAPI stain for the mutant TBS? It should stain, as it is still DNA. Also, why is there no LLPS despite it being a mutant piece of DNA, as TCP22 and CRY2 can phase separate in vitro both separately and together in the absence of DNA. Also, an important control would be to use a random piece of DNA that has nothing to do with TBS to test for phase separation. This will further show that this is a sequence specific event vs something that can be induced merely through electrostatic interactions.

For figure 1 (lines 130-147), the authors do not clarify if they are using WT protoplasts or mutant protoplasts. If WT, should there be a difference in photobody morphology as there should be endogenous TCP22? Please have the authors specify the genetic background to their experiments. The lack of genotype clarification isn't unique to figure 1. Please go through all figures and make sure the genetic background is stated for each experiment.

In general for figure 1, the authors make a lot of hand wave-like conclusions without proper evidence. For instance, are the slower recovery rate seen in figure 1D actually due to slower recovery or due to lower starting concentrations of CRY (are these values normalized by total concentration?). Also, the results in figure 1D are not consistent with the FRAP experiment from supplemental figure 4B, which also looks at CRY. Then they also make a lot of comments about the potential role of the CRY IDR but do not actually test the role (see major comment 1).

For supplementary figure 2, they switch to using a 488nm laser, which is more appropriate for blue-light studies. For supplementary figure 2 C in particular, they say that TCP22 alone cannot phase separation in vivo. Although I do think their results are likely correct, they do not quantify the intensity of TCP22 nuclear fluorescence. Does it have the same concentration as CRY2? As the formation of photobodies is concentration dependent, it is important to know whether the lack of phase separation is due to lower concentrations during transfection. This seems to be a common problem within their manuscript in general, as they are using OX constructs. In experiments where phase separation is not occurring, please quantify the fluorescent intensity against the sample that is phase separating.

For figure 2 and other figures where you use cry1/cry2 and tcp20/tcp22 mutants, please explain why you are using double mutants versus single mutants. Could CRY1 or TCP20 also influence LLPS or circadian responses?

Reviewer #3 (Remarks to the Author):

Cryptochromes are blue light receptors that also function as a component of the central clock oscillator. This work focuses on the CRY2 cryptochrome from Arabidopsis, demonstrating that this protein undergoes blue-light-dependent condensation, likely by liquid-liquid phase separation.

This work makes several exciting claims, including that CRY2 phase separation allows for CRY2 to act as a scaffold to recruit multiple clients to regulate a biological process. If true, this would be a big step forward in our understanding of the integration of blue light signaling and the clock. However, weaknesses in the data would need to be addressed before these claims can be fully supported.

There are several major limitations to this work:

- 1- Nearly all data are collected using a transient overexpression system (Arabidopsis protoplasts) in which protein levels accumulate to much higher levels than what are physiologically relevant. LLPS is dependent on protein concentration; it is unclear from this data whether any of these protein behaviors occur at native protein concentrations.
- 2- For several experiments, the data is further complicated by the use of split YFP to examine phase separation – using this system rather than separately tagged proteins complicates our ability to interpret these results. Members of the LLPS community are often cautioned from using FPs that self-dimerize. In this case, the authors use a split FP that is known to stabilize transient interactions.
- 3- The authors overinterpret much of their work (see below).

Unsupported claims:

“of CRY2-dependent condensate; subsequently, the accumulation of the IDR (Intrinsic Disorder Region) of CRY2 and CRY2-interacting protein (e.g. TCP22) drives these proteins into phase-separation in a condensed state in plant cells.” There is no evidence that the IDRs of these proteins play any role in this behavior.

“This indicates that CRY2-only condensates have liquid-like properties, but the liquid is probably viscous; it is likely on the boundary between the liquid-liquid phase and gel phase, which is possibly caused by the short IDR (Intrinsic Disorder Region) of CRY2 (Supplementary Fig. 5c).” Again, there is insufficient data to support any role for the CRY2 IDR in this behavior.

“This indicates that CRY2-only condensates have liquid-like properties, but the liquid is probably viscous; it is likely on the boundary between the liquid-liquid phase and gel phase, which is possibly caused by the short IDR (Intrinsic Disorder Region) of CRY2 (Supplementary Fig. 5c).” Again, no data to support this claim.

“CRY2 alone is sufficient to form the condensates in cells after blue light irradiation (Supplementary Fig. 4a).” This conclusion cannot be drawn, as the experiment was done in plant cells with a host of various proteins present.

Other comments

- I don't understand why the authors discuss their data in the Introduction. All data should be included in results.
- Many experiments are not described in sufficient detail in the text or figure legend to figure out exactly what was done. More detail needs to be added.
- The CRY2 D387A mutation was not explained until well after the first experiments using this variant.

REVIEWER COMMENTS

Reviewer #1 (Remarks to the Author):

The current manuscript explores an important question with respect to light inputs into the circadian clock in *Arabidopsis*, namely what is the role of the cryptochromes in this response at the molecular level; this is doubly important since cryptochromes play important roles in the circadian clocks of animals and other organisms. The authors provide evidence of an intriguing novel mechanism in which a combination of Cry2 and transcription factor TCP22 undergoes selective condensation in blue light, further recruiting additional proteins to interact with the oscillator component CCA to alter circadian rhythms in blue light. A particularly elegant feature of this study is that the results were replicated in a recombinant mammalian expression system. Because this blue-light dependent condensation does not require blue-light dependent interaction, but only small conformational change of constitutive complexes, it represents a novel means whereby photoreceptors may provide light input into targeted processes. The data and controls are convincing and well presented. For all these reasons, the study merits publication and is likely of interest to a wide audience.

However, a point which the authors largely gloss over is that cryptochromes, and certainly cry2, do not play any major role in the plant circadian clock, and in fact have even been reported to play light-independent functions in red light and affect features such as amplitude and period length - which definitely do not fit into the present scheme (see Devlin and Kay, *Plant Cell*, 2000). Cry2 in particular is unlikely to play any significant role in blue light, as it is rapidly degraded and quite simply disappears. Apart from that, dozens of articles show rhythmicity in plants in the absence of crys. So, the statement of the authors implying cryptochromes are the major receptor implicated in plant circadian clock need to be softened. A better approach would be to claim an impact on the clock, along with many other players, and speculate for instance how recruitment of pifs and phytochromes to the condensed phase could provide an additional mechanism of cry role in the plant clock.

Some minor points - one is puzzled why there is so little data regarding Cry1, especially as it has been shown to function redundantly with cry2 in the clock. The authors should at least point out parallels (in mechanism of phosphorylation and interaction with similar partners) in comparison to cry2 function

Also, there are many typos and spelling errors, even in the abstract where the authors manage to mis-spell cryptochrome. This should be attended to in a revision.

Reviewer #2 (Remarks to the Author):

This paper aims to identify the contributions of TCP22 and CRY2 in mediating the circadian clock under blue-light conditions. This research question is of interest because TCP22 and CRY2 have been previously shown to constitutively interact with each other despite the presence or absence of blue light signals. Through the emergence of the LLPS field, the authors hypothesize that LLPS could be used as a biophysical mechanism that facilitates

TCP22-CRY2 blue light responses, without the need for TCP22 and CRY2 to physically disassociate. Using a mix of biochemistry, confocal imaging, and proteomics, the authors impressively show that TCP22 and CRY2 phase separate to form photobodies within protoplast cells, and that these photobodies have the ability to regulate transcription of the circadian clock protein CCA1. Additionally, they find that two other proteins, kinase PPK1 and light regulated LWD1, assemble into photobodies and also enable blue light responses by helping with both phosphorylation and transcriptional efficiency of TCP22-CRY2 photobodies. There are many nice things about this paper however, more experiments and clarifications must be made before it should be published. Additionally, the authors may want to have someone proof read it for grammatical errors and sentence structure.

Comments:

Lines 52—55: The authors mention that the interaction of nCRY2 and cTCP22 formed fluorescent photobodies in response to blue light (figure 1d). However, they use a 514 nm laser, which illuminates green light. Will the authors please explain why they did not use a 488nm laser for excitation.

Lines 82-87: The authors conclude that CRY2-TCP22 LLPS is blue-light specific. However, they never directly test the formation of CRY2-TCP22 condensates under other light sources. To me, this suggests that blue light is sufficient for LLPS but does not rule out the possibility of other light sources triggering their formation. Please have the authors confirm that this is a blue-light specific response.

Lines 122-128: The author's hypothesis for the role of CRY2's IDR is interesting and should be tested. Since they are doing this work in protoplasts, the authors should test how truncation or modifications to the IDR region influence phase separation.

Lines 173-186: To tie the relationship between CRY2-TCP22 condensates and CCA1 transcription together, the authors look to see if the TBS motif of the CCA1 promoter can trigger LLPS in vitro. However, this experiment is flawed/ missing some vital controls. For instance, for supplemental figure 6, why is there no DAPI stain for the mutant TBS? It should stain, as it is still DNA. Also, why is there no LLPS despite it being a mutant piece of DNA, as TCP22 and CRY2 can phase separate in vitro both separately and together in the absence of DNA. Also, an important control would be to use a random piece of DNA that has nothing to do with TBS to test for phase separation. This will further show that this is a sequence specific event vs something that can be induced merely through electrostatic interactions.

For figure 1 (lines 130-147), the authors do not clarify if they are using WT protoplasts or mutant protoplasts. If WT, should there be a difference in photobody morphology as there should be endogenous TCP22? Please have the authors specify the genetic background to their experiments. The lack of genotype clarification isn't unique to figure 1. Please go through all figures and make sure the genetic background is stated for each experiment.

In general for figure 1, the authors make a lot of hand wave-like conclusions without proper evidence. For instance, are the slower recovery rate seen in figure 1D actually due to slower

recovery or due to lower starting concentrations of CRY (are these values normalized by total concentration?). Also, the results in figure 1D are not consistent with the FRAP experiment from supplemental figure 4B, which also looks at CRY. Then they also make a lot of comments about the potential role of the CRY IDR but do not actually test the role (see major comment 1).

For supplementary figure 2, they switch to using a 488nm laser, which is more appropriate for blue-light studies. For supplementary figure 2 C in particular, they say that TCP22 alone cannot phase separate in vivo. Although I do think their results are likely correct, they do not quantify the intensity of TCP22 nuclear fluorescence. Does it have the same concentration as CRY2? As the formation of photobodies is concentration dependent, it is important to know whether the lack of phase separation is due to lower concentrations during transfection. This seems to be a common problem within their manuscript in general, as they are using OX constructs. In experiments where phase separation is not occurring, please quantify the fluorescent intensity against the sample that is phase separating.

For figure 2 and other figures where you use cry1/cry2 and tcp20/tcp22 mutants, please explain why you are using double mutants versus single mutants. Could CRY1 or TCP20 also influence LLPS or circadian responses?

Reviewer #3 (Remarks to the Author):

Cryptochromes are blue light receptors that also function as a component of the central clock oscillator. This work focuses on the CRY2 cryptochrome from Arabidopsis, demonstrating that this protein undergoes blue-light-dependent condensation, likely by liquid-liquid phase separation.

This work makes several exciting claims, including that CRY2 phase separation allows for CRY2 to act as a scaffold to recruit multiple clients to regulate a biological process. If true, this would be a big step forward in our understanding of the integration of blue light signaling and the clock. However, weaknesses in the data would need to be addressed before these claims can be fully supported.

There are several major limitations to this work:

1- Nearly all data are collected using a transient overexpression system (Arabidopsis protoplasts) in which protein levels accumulate to much higher levels than what are physiologically relevant. LLPS is dependent on protein concentration; it is unclear from this data whether any of these protein behaviors occur at native protein concentrations.

2- For several experiments, the data is further complicated by the use of split YFP to examine phase separation – using this system rather than separately tagged proteins complicates our ability to interpret these results. Members of the LLPS community are often cautioned from using FPs that self-dimerize. In this case, the authors use a split FP that is known to stabilize transient interactions.

3- The authors overinterpret much of their work (see below).

Unsupported claims:

“of CRY2-dependent condensate; subsequently, the accumulation of the IDR (Intrinsic Disorder Region) of CRY2 and CRY2-interacting protein (e.g. TCP22) drives these proteins into phase-separation in a condensed state in plant cells.” There is no evidence that the IDRs of these proteins play any role in this behavior.

“This indicates that CRY2-only condensates have liquid-like properties, but the liquid is probably viscous; it is likely on the boundary between the liquid-liquid phase and gel phase, which is possibly caused by the short IDR (Intrinsic Disorder Region) of CRY2 (Supplementary Fig. 5c).” Again, there is insufficient data to support any role for the CRY2 IDR in this behavior.

“This indicates that CRY2-only condensates have liquid-like properties, but the liquid is probably viscous; it is likely on the boundary between the liquid-liquid phase and gel phase, which is possibly caused by the short IDR (Intrinsic Disorder Region) of CRY2 (Supplementary Fig. 5c).” Again, no data to support this claim.

“CRY2 alone is sufficient to form the condensates in cells after blue light irradiation (Supplementary Fig. 4a).” This conclusion cannot be drawn, as the experiment was done in plant cells with a host of various proteins present.

Other comments

- I don't understand why the authors discuss their data in the Introduction. All data should be included in results.

- Many experiments are not described in sufficient detail in the text or figure legend to figure out exactly what was done. More detail needs to be added.

- The CRY2 D387A mutation was not explained until well after the first experiments using this variant.

Responses to the reviewers' comments

Notes:

1. The reviewers' comments are underlined, the authors' revised text and newly added figures are highlighted by the blue color, which are also searchable in revised manuscript to find their location in the revision.

Responses to Review #1

1. However, a point which the authors largely gloss over is that cryptochromes, and certainly cry2, do not play any major role in the plant circadian clock, and in fact have even be reported to play light-independent functions in red light and affect features such as amplitude and period length - which definitely do not fit into the present scheme (see Devlin and Kay, Plant Cell, 2000). Cry2 in particular is unlikely to play any significant role in blue light, as it is rapidly degraded and quite simply disappears. Apart from that , dozens of articles show rhythmicity in plants in the absence of crys. So, the statement of the authors implying cryptochromes are the major receptor implicated in plant circadian clock need to be softened. A better approach would be to claim an impact on the clock, along with many other players, and speculate for instance how recruitment of pifs and phytochromes to the condensed phase could provide an additional mechanism of cry role in the plant clock.

RESPONSE: We agree. We softened the description “cryptochromes are the major receptor implicated in plant circadian clock” and modified the sentence in the revision to address this issue: “We observed *CCA1* was arrhythmic in *cry1cry2* background under blue light, which suggested that **cryptochromes are the important circadian photoreceptors along with their interaction proteins together to mediate blue light input and calibrate the rhythm of core oscillator component *CCA1*.**” We also modified this sentence to illustrate the efforts of PIFs and phytochromes: “**For example, the PIFs, even phytochromes may be involved in the phase-separation responses to regulate the blue light input to the clock, as they are also photobody proteins interacting with cryptochromes and exhibiting the function of regulating circadian clock in blue light.**^{42-44.}”

We also appreciate the reviewer's suggestion about whether cryptochromes play the major role in blue light input to the circadian clock. We have comprehended that the circadian clock is still rhythmic in plants in the absence of cryptochromes from many studies. We have also learned the cryptochrome mutants are still rhythmic, and they affect the features such as amplitude and period length in the previous study (Devlin and Kay, Plant Cell, 2000). However, those studies mainly focus on the output of clock and check the LUC activities with

CCR or *CAB* promoter. Different from those papers, this study only focuses on the process of blue light input to the clock, but not the output of the clock and mainly checks the LUC activities with *CCAI* promoter. As we know, the input of the clock is different from the output of the clock, which means CRYs could take a major role in the process of blue light input to the clock but not output. Because many proteins (e.g. ZTLs, ELF3, PIFs and phytochromes) could neutralize the effort of CRYs during the clock running, which represents a different output signal of the clock in blue light (Sanchez et al., 2020). Since this study focuses on the process of blue light input to the clock but not the output, we have tried many times about the result of Fig. 2f, and confirmed that the transcription of *CCAI* is really arrhythmic in *cry1cry2* under blue light. Even the single mutant *cry2* is enough to regulate the amplitude and the period length of *CCAI* (New Figure R1), which indeed suggests that CRYs are the major photoreceptors for blue light to input signal to the core oscillator component *CCAI* of circadian clock. And we also trusted that the output of the circadian clock (e.g. *CAB2::LUC*) is still rhythmic in the same condition, just like the description in Devlin and Kay, Plant Cell, 2000.

We really thank the reviewer for raising the interesting issue. Although a meaningful discussion of this important but complex issue is out of the scope of this study, we are presently preparing a separate manuscript to discuss in more details about cryptochromes regulating the relationship between the input and output of the clock in blue light.

New Figure R1. Bioluminescence analysis showed amplitude and rhythm change of *CCAI::LUC* expression in indicated genotypes. WT, *cry2* seedlings ($n \geq 30$) expressing *CCAI::LUC* were grown on 1/2 MS solid medium in 12 h dark/12 h light ($75 \mu\text{mol m}^{-2} \text{s}^{-1}$ white light) treatment for 3 days. These seedlings then transferred into continuous blue light ($5 \mu\text{mol m}^{-2} \text{s}^{-1}$) for 4 days, subsequently, LUC activity was detected for luciferase activity under continuous blue light ($5 \mu\text{mol m}^{-2} \text{s}^{-1}$). Data are presented as mean \pm SD ($n = 3$).

2. one is puzzled why there is so little data regarding Cry1, especially as it has been shown to function redundantly with cry2 in the clock. The authors should at least point out parallels (in mechanism of phosphorylation and interaction with similar partners) in comparison to cry2 function

RESPONSE: we thank the reviewer for raising the nice issue. Actually, we have analyzed the phase-separation state of CRY1 in blue light since we started the investigation on how phase-separation regulates the circadian clock. However, CRY1 itself didn't exhibit the phase-separation changes in blue light, which is different from CRY2 (New Figure R2). Additionally, CRY1 could form the photobodies with SPA1/COP1(Liu et al., 2011; Van Buskirk et al., 2012) that it seems CRY1 possesses a different way for phase separation. Since CRY1 is also involved in photobody formation with other proteins out of the nucleus (New Figure R2), it is more complicated than CRY2 processing phase-separation merely in nucleus. It is difficult to discuss the phase-separation mechanism of CRY1 and CRY2 clearly in a single manuscript. We have started the study of CRY1 phase-separation for another paper.

New Figure R2. Images of CRY1-nYFP/CRY1-cYFP, CRY1-GFP (a) , CRY1-nYFP/ProteinX-cYFP (b) under laser treatment. Indicated plasmids were transfected into 3-week-old wide type (Col4) Arabidopsis protoplasts. 488 nm laser (for GFP) or 514 nm (for YFP) laser of confocal microscope was used to capture image and treated the protoplasts.

3. there are many typos and spelling errors, even in the abstract where the authors manage to mis-spell cryptochrome. This should be attended to in a revision.

RESPONSE: We thank the reviewer's suggestion. We have checked the manuscript carefully, and revised 85 issues of solecism typos and spelling errors.

Responses to Review #2

1. Lines 52—55: The authors mention that the interaction of nCRY2 and cTCP22 formed fluorescent photobodies in response to blue light (figure 1d). However, they use a 514 nm

laser, which illuminates green light. Will the authors please explain why they did not use a 488nm laser for excitation.

RESPONSE: Since most fluorescent photobodies are the YFP-fusing proteins, and the excitation wavelength of YFP is 514 nm, we chose 514nm to excite YFP but not 488nm. But we have also tried to use 488nm to excite the phase-separation of CRY2-TCP22, and then to excite the YFP at 514nm. Although with different photoabsorption intensity, it exhibited the similar results with those assay only using 514nm laser. We have also examined the light spectrum, and the 514nm is the cyan light mixing green light and blue light together, which could photo-excite CRY2 and form the condensates.

2. Lines 82-87: The authors conclude that CRY2-TCP22 LLPS is blue-light specific. However, they never directly test the formation of CRY2-TCP22 condensates under other light sources. To me, this suggests that blue light is sufficient for LLPS but does not rule out the possibility of other light sources triggering their formation. Please have the authors confirm that this is a blue-light specific response.

RESPONSE: We agree. We have shown that CRY2 mainly photoexcited and interacted with other proteins in blue light previously (Liu et al., 2008; Liu et al., 2017; Wang et al., 2016; Zuo et al., 2011), and we usually analyzed the behaviors of CRY2 and its interaction proteins in dark (0s) and various time points of the blue light irradiation in our previous studies. But we thank the reviewer for raising this question this time. We have analyzed the CRY2-TCP22 condensates under various wavelength lights and added the new figure in this revision. Newly added Supplementary Fig. 2 exhibited that CRY2 and TCP22 could form the phase separation under blue light (488nm) and cyan light composed of blue light and green light (514nm and 530 nm). CRY2-TCP22 LLPS exhibited the blue light specificity in various wavelengths of cyan light, but the formation of CRY2-TCP22 LLPS under 530 cyan light was much weaker than that under 514nm cyan light containing more blue light. More importantly, CRY2 and TCP22 can't form the condensates in the pure green light (550nm) and the light possessing wavelength longer than 550nm. These data confirmed that CRY2-TCP22 LLPS is blue-light specific. We also revise the sentence in text: “Not only the formation, but the size and the liquid properties of the CRY2-TCP22 condensates depend on the irradiation time, the fluence rate of blue light and the wavelength of light (Supplementary Fig. 1c-g and Supplementary Fig. 2)”

3. Lines 122-128: The author's hypothesis for the role of CRY2's IDR is interesting and should be tested. Since they are doing this work in protoplasts, the authors should test how truncation or modifications to the IDR region influence phase separation.

RESPONSE: We agree. And we thank the reviewer for pointing out this relevant issue that we had previously overlooked. We have added the truncation protein for phase-separation assay (Supplementary Fig. 9a-d) and added the text that “We truncated the IDR³² (Intrinsic Disorder Region) domain of CRY2 (CRY2^{ΔIDR}) and compared the dynamic responses of CRY2^{ΔIDR}-TCP22 condensates with those of CRY2-TCP22 condensates. Without the short IDR of CRY2, the TCP22- CRY2^{ΔIDR} condensates significantly decreased the dynamic compared with CRY2-TCP22 condensates (Supplementary Fig. 9a, b). Additionally, since the TCP22 possesses a longer IDR domain than CRY2 (Supplementary Fig. 9e, f), the TCP22-CRY2^{ΔIDR} condensates were more dynamic than CRY2-CRY2^{ΔIDR} condensates with a shorter IDR domain (Supplementary Fig. 9c, d).”

4. Lines 173-186: To tie the relationship between CRY2-TCP22 condensates and CCA1 transcription together, the authors look to see if the TBS motif of the CCA1 promoter can trigger LLPS in vitro. However, this experiment is flawed/ missing some vital controls. For instance, for supplemental figure 6, why is there no DAPI stain for the mutant TBS? It should stain, as it is still DNA. Also, why is there no LLPS despite it being a mutant piece of DNA, as TCP22 and CRY2 can phase separate in vitro both separately and together in the absence of DNA.

RESPONSE: “For instance, for supplemental figure 6, why is there no DAPI stain for the mutant TBS?” The mutant TBS can not form the DAPI stain with CRY2-TCP22 below the protein concentration threshold that CRY2 and TCP22 can't form the condensates. Since the mutant TBS could not promote CRY2 to form condensates with TCP22, but the native TBS could promote CRY2-TCP22 condensates, we could only detect the CRY2-TCP22-native TBS condensates (also the DAPI stain of native TBS) at the protein concentration, in which CRY2 and TCP22 can't form the condensates by themselves (6 μM TCP22 and 1 μM CRY2). We have exhibited the protein concentration threshold of CRY2-TCP22 condensate formation in Fig. 1k previously. In this revision, we also added the newly figure (Supplementary Fig. 11e) to show that *in vitro*, the CRY2-TCP22 can't form the droplet at that protein concentration (6 μM TCP22 and 1 μM CRY2).

Since the mutant TBS is still DNA, we could see the smear blue all over the visual field with mutant TBS sample in DAPI channel, but not the DAPI stain. Because the mutant TBS can't form the droplet with CRY2 and TCP22 at that protein concentration (6 μM TCP22 and 1 μM CRY2).

“Also, why is there no LLPS despite it being a mutant piece of DNA, as TCP22 and CRY2 can phase separate in vitro both separately and together in the absence of DNA.” CRY2 and TCP22 could form LLPS *in vitro*, but the protein concentration should be beyond a threshold. Since the concentration (6 μ M TCP22 and 1 μ M CRY2) is under the threshold (we have previously exhibited the threshold issue in Fig. 1k), they couldn't form LLPS in this protein concentration by themselves or with the mutant TBS. However, since the native TBS promotes the formation of CRY2-TCP22 LLPS, we could see the CRY2-TCP22 LLPS in such protein concentration with native TBS. To further exhibit the threshold of CRY2-TCP22 condensates of this issue, we revised the previous supplementary figure 6e (it became the Supplementary Fig. 11e in the revised version), and added the negative control of 6 μ M TCP22 and 1 μ M CRY2, which showed no CRY2-TCP22 droplet.

“Also, an important control would be to use a random piece of DNA that has nothing to do with TBS to test for phase separation. This will further show that this is a sequence specific event vs something that can be induced merely through electrostatic interactions.”

We thank the suggestion of the reviewer. We added the control of random piece of DNA (random DNA with 6 μ M TCP22 and 1 μ M CRY2), which exhibited no CRY2-TCP22 droplet either. We revised the previous figure and the sentence: “Importantly, only the TBS motif^{33,34} of the CCAI promoter but not a mutant TBS motif or random piece of DNA could enhance the CRY2-TCP22 LLPS, which revealed that the DNA accelerated the CRY2-TCP22 LLPS in a DNA sequence dependent manner (Supplementary Fig. 11e).”

5. For figure 1 (lines 130-147), the authors do not clarify if they are using WT protoplasts or mutant protoplasts. If WT, should there be a difference in photobody morphology as there should be endogenous TCP22? Please have the authors specify the genetic background to their experiments. The lack of genotype clarification isn't unique to figure 1. Please go through all figures and make sure the genetic background is stated for each experiment.

RESPONSE: We have used the WT protoplasts in Figure 1, and we have also revised the figure legend for all figures referred to this issue. Yes, WT protoplasts contain the endogenous TCP22. But the endogenous TCP22 is much less than the overexpressed YC-TCP22, so in the WT protoplasts, the YCTCP22-YNCRY2 condensates still exhibited more dynamic than YCCRY2-YNCRY2 condensates in figure 1.

And we appreciate the reviewer for raising this issue. To completely solve this issue in the revision, we compared the CRY2 photobody morphology in *Myc-TCP22*, WT and *tcp22tcp20* double mutant (Supplementary Fig. 4a), which contain the similar amount of CRY2 protein (Supplementary Fig. 4b). As shown in Supplementary Fig. 4a, in Col-4, the endogenous CRY2 and TCP22 could form several condensates compared with *tcp20tcp22* lines without any condensates. However, in the TCP22-overexpression lines, the CRY2/TCP22 condensates significantly increased. These results confirmed our previous suggestion that TCP22 could enhance the formation of CRY2-related condensates in plant cells. Without TCP22 antibody, we further expressed Myc-TCP22 driven by native promoter in *tcp20tcp22* lines to examine the condensates formation of TCP22. We added the new data and a sentence to explain: “This suggestion was further confirmed by the immunostaining assay, in which CRY2 only formed the condensates with Myc-TCP22 driven by native promoter in blue light, in contrast, no condensate was exhibited in *tcp20tcp22* mutant or in red light (Supplementary Fig. 10 a, b).”

6. In general for figure 1, the authors make a lot of hand wave-like conclusions without proper evidence. For instance, are the slower recovery rate seen in figure 1D actually due to slower recovery or due to lower starting concentrations of CRY (are these values normalized by total concentration?).

RESPONSE: Since we lacked the TCP22 antibody or split YFP antibody, we carefully analyzed the protein concentration with fluorescence intensity of measured cells (Cho et al., 2004; Laufer et al., 2019; Xu et al., 2017), and compared the recovery rate of different cells with the similar fluorescence intensity previously. In this revision, we repeated this assay and added the Supplementary Figure 1b to exhibit the similar fluorescence intensity of different cells we compared. To further confirm the CRY2 concentration in all newly added immunostaining assay, we examined the CRY2 concentration in the newly added data (Supplementary Fig. 4 and Supplementary Fig. 10) by western blot, which exhibited that the CRY2 concentration can't be affected by TCP22 in WT and *tcp22tcp20* double mutant (Supplementary Fig. 4b, d and Supplementary Fig. 10b), and TCP22 could promote the CRY2 condensate formation in plant cells (Supplementary Fig. 4a, c and Supplementary Fig. 10a).

Also, the results in figure 1D are not consistent with the FRAP experiment from supplemental figure 4B, which also looks at CRY. Then they also make a lot of comments about the potential role of the CRY IDR but do not actually test the role (see major comment 1).

RESPONSE: We are sorry, but we don't understand about this comparison raised by the reviewer. The Figure 1d shows the condensate formation of the whole cell in blue light, but the supplementary Figure 4b (it became the Supplementary Fig. 7b in the revised version) is about the FRAP assay showing the recovery rate of the single condensate but not the whole cell. Yes, both of them are about CRY, but Figure 1d and supplementary Figure 4b show different objects (one is about the whole cell and the other the single condensate) and different assays (fluorescence observation and belching-recovering), respectively.

If the reviewer means to compare Figure 1d and supplementary Figure 4a(it became the Supplementary Fig. 7a in the revised version), we could answer that, yes, they are still different although both of them look at CRY2. Because the YFP and 514nm laser were used for Figure 1d, but GFP and 488nm laser were used for supplementary Figure 4a, thus the CRY2-related condensates in 488nm were stronger than those in 514nm. The newly added supplementary Figure 2 could also exhibit this. In any case, both Figure 1d and supplementary Figure 4b suggest the same issue that blue light enhanced the photobody formation of CRY2.

7. For supplementary figure 2, they switch to using a 488nm laser, which is more appropriate for blue-light studies.

RESPONSE: We switched to use 488nm laser for Fig. 2, because we needed to use 488nm laser to excite the GFP to observe the CRY2-GFP. At the beginning of this study, we identified with the reviewer that 488nm laser is more appropriate for blue light studies. But later, we found that CRY2 is sensitive to blue light, the intensity of 488nm laser is too strong for CRY2, and the process of CRY2 photoexciting and photobody formation accomplished very fast that it was hard to examine. So in the main text, we usually used YFP [being excited by cyan light (514nm) and in lower blue light intensity] to analyze the photobody of CRY2 and related proteins.

For supplementary figure 2 C in particular, they say that TCP22 alone cannot phase separation in vivo. Although I do think their results are likely correct, they do not quantify the intensity of TCP22 nuclear fluorescence. Does it have the same concentration as CRY2? As the formation of photobodies is concentration dependent, it is important to know whether the lack of phase separation is due to lower concentrations during transfection. This seems to be a common problem within their manuscript in general, as they are using OX constructs. In experiments where phase separation is not occurring, please quantify the fluorescent intensity against the sample that is phase separating.

RESPONSE: We thank the reviewer for raising this issue. The 35s overexpression may explain this issue as discussed above, but not rigorous enough. So in this revision, we added the new data to confirm this. In the newly added supplementary Fig. 4a, we transferred the *Myc-TCP22* into WT and *cry1cry2* mutant. Using immunostaining assay, we could detect the Myc-TCP22 protein (smear in cells) in *cry1cry2* mutant, but we could only observe the photobody of Myc-TCP22 in wild type (Col-4) but not in *cry1cry2* background. The western blot also indicated the similar concentration of CRY2 and TCP22 (The phosphorylation TCP22 could increase under blue light in WT but not in *cry1cry2*, which is similar to the description of the phosphorylation part) (supplementary Fig. 4b). These results further suggested that TCP22 alone cannot form the phase separation *in vivo*.

8. For figure 2 and other figures where you use *cry1/cry2* and *tcp20/tcp22* mutants, please explain why you are using double mutants versus single mutants. Could CRY1 or TCP20 also influence LLPS or circadian responses?

RESPONSE: Previous studies suggest that both CRY1 and TCP20 are redundant to CRY2 and TCP22 in circadian clock regulation, respectively (Devlin and Kay, 2000; Wu et al., 2016). And in this study, we mainly discussed the regulation of CRY2 and TCP22 to the circadian clock in Figure 2. To eliminate the interferes of CRY1 and TCP20 in circadian assay, we used *cry1/cry2* and *tcp20/tcp22* mutants as background, and expressed CRY2 and TCP22 to evaluate the circadian function of those proteins.

Could CRY1 or TCP20 also influence LLPS or circadian responses? This is an interesting question that we do not have an answer yet. Under our investigation, different from TCP22, TCP20 can't be phosphorylated by PPKs, and CRY1 exhibited no phase separation state in blue light. Although TCP20 and CRY1 were involved in the regulation of circadian clock (Somers et al., 1998; Wu et al., 2016), but it is implied that TCP20 and CRY1 may possess more complicated pathway to input their signal to the circadian clock, which should be investigated in future studies.

Responses to Review #3

1. Nearly all data are collected using a transient overexpression system (Arabidopsis protoplasts) in which protein levels accumulate to much higher levels than what are physiologically relevant. LLPS is dependent on protein concentration; it is unclear from this data whether any of these protein behaviors occur at native protein concentrations.

RESPONSE: we agree, and we thank the reviewer for raising the important issue of this study. The overexpression issue is mostly due to historical and technical reasons in my lab. We usually use 35S to observe the phenomena of protein in cells or protoplasts, because the increasing amount of the target protein usually eliminates the interference from other proteins which also interact with the target protein. However, the overexpression protein may affect the relevant physiological process as suggested by the reviewer. We added the new results to address this overexpression issue. The newly added Supplementary Fig. 3a showed that CRY2 driven by its own native promoter also could form the condensates in wild-type plants. And the number of condensates increased after a longer blue light irradiation. Additionally, the newly added Supplementary Fig. 3c and Supplementary Fig. 10a also exhibited that cYFP-TCP22 or Myc-TCP22 driven by its native promoter formed the condensates with CRY2 or nYFP-CRY2 driven by CRY2 native promoter in a blue light dependent manner. In the new version, we added the new sentences and figures to address this issue: “Additionally, we further confirmed that the native CRY2 formed the condensates in the wild type plant cell utilizing the immunostaining assay (Supplementary Fig. 3a-b); and CRY2-TCP22 also formed the condensates in a blue light dependent manner when CRY2 and TCP22 expressed under their native promoters, respectively (Supplementary Fig. 3c). To further eliminate the interference of split fluorescent protein on CRY2-TCP22 phase separation, the CRY2-TCP22 condensates were analyzed in wild-type or transgene plants with immunostaining assay. As shown in Supplementary Fig. 4a-b, without YFP fusion domain, the native CRY2 formed the condensates in a blue light dependent manner, and Myc-TCP22 also formed the blue light specific condensates with CRY2 and promoted the condensate formation in blue light (Supplementary Fig. 4c-d).” “This suggestion was further confirmed by the immunostaining assay, in which CRY2 only formed the condensates with TCP22 driven by native promoter in blue light, in contrast, no condensate was exhibited in *tcp20tcp22* mutant or in red light (Supplementary Fig.10 a, b).”

2. For several experiments, the data is further complicated by the use of split YFP to examine phase separation – using this system rather than separately tagged proteins complicates our ability to interpret these results. Members of the LLPS community are often cautioned from using FPs that self-dimerize. In this case, the authors use a split FP that is known to stabilize transient interactions.

RESPONSE: Actually, we prepared the GFP tagged TCP22 for the LLPS assay at the very beginning. Unfortunately, both the N and C-terminal GFP-tag TCP22 degraded in plant cells (Actually, we tried to fuse more FPs than to only use GFP) (New Figure R3), and this phenomenon also occurred with other FP fusion

proteins containing the IDR domain (Arribas-Hernández et al., 2021). Luckily, the split YFP fusion TCP22 didn't.

We appreciate that the reviewer raised the issue of fluorescent protein in phase separation assay. Most previous studies of phase separation even utilized the fluorescent protein as a tool to investigate the biochemical or physiological function of proteins involving LLPS (Fang et al., 2019; Jung et al., 2020; Zavaliev et al., 2020). We have also noticed that the fluorescent proteins need to be handled in phase separation assay carefully. We checked whether the split YFP interferes the LLPS condensates formation of CRY2 and TCP22. As shown in New Figure R4, neither N nor C-split YFP itself could form the condensate with split-YFP-fused CRY2 or TCP22. Nevertheless, we understood that this result still couldn't fully confirm whether the split YFP stabilizes the CRY2-TCP22 condensates. We thus added the immunostaining to analyze whether the native CRY2 and TCP22 could blue light specifically form the condensates. Since we utilized the CRY2-antibody and CY3, 488 dye to detect the protein, this assay eliminated the interference of FPs or split FPs. As shown in the newly added Supplementary Fig. 4 and Supplementary Fig. 10, the immunostaining assay confirmed our previous split-YFP assay, and exhibited that the native CRY2 in wild type (Col-4) formed the condensates in a blue light dependent manner. Since we failed to prepare the TCP22 antibody, we transferred Myc-TCP22 into Col-4, *cry1cry2* and *tcp20tcp22*, respectively. We co-detected the Myc-TCP22 with Myc antibody and 488 dye, which suggested that the CRY2-TCP22 also formed the blue light specific condensates. Furthermore, CRY2-TCP22 exhibited faster formation and bigger size than CRY2-only condensates. We revised the manuscript with the following description and figures to confirm that the CRY2-TCP22 condensates could also occur in wild type: “To further eliminate the interference of split fluorescent protein on CRY2-TCP22 phase separation, the CRY2-TCP22 condensates were analyzed in wild-type plants with immunostaining assay. As shown in Supplementary Fig. 4a-b, without YFP fusion domain, the native CRY2 formed the condensates in a blue light dependent manner, and Myc-TCP22 also formed the blue light specific condensates with CRY2 to promote the condensate formation in blue light (Supplementary Fig. 4c-d).” Here, we thank the reviewer again for pointing out this relevant issue that enable our manuscript to be more rigorous, although most previous studies of phase separation haven't carefully analyzed this issue.

New Figure R3. Degradation of GFP tagged TCP22 in Arabidopsis.

Five-day-old seedlings overexpressing GFP-TCP22, TCP22-GFP and GFP were lysed and then detected by western blot. Indicated proteins were detected by anti-GFP.

New Figure R4. Neither N nor C-split YFP itself could form the condensate with split-YFP-fused CRY2 or TCP22. Indicated plasmids were transfected into protoplasts isolated from 3-week-old Arabidopsis (Col4). The fluorescence signals were detected using 514nm laser. Scale bar, 50 μ m.

3. of CRY2-dependent condensate; subsequently, the accumulation of the IDR (Intrinsic Disorder Region) of CRY2 and CRY2-interacting protein (e.g. TCP22) drives these proteins into phase-separation in a condensed state in plant cells.” There is no evidence that the IDRs of these proteins play any role in this behavior

RESPONSE: We agree. We revised the manuscript that: “subsequently, CRY2 recruits its interacting proteins (e.g. TCP22) and drives these proteins into phase-separation in a condensed state in plant cells.”

4. This indicates that CRY2-only condensates have liquid-like properties, but the liquid is probably viscous; it is likely on the boundary between the liquid-liquid phase and gel phase, which is possibly caused by the short IDR (Intrinsic Disorder Region) of CRY2 (Supplementary Fig. 5c).” Again, there is insufficient data to support any role for the CRY2 IDR in this behavior.

RESPONSE: We agree. In this revision, we truncated CRY2-IDR, and exhibited the different efforts between the short IDR of CRY2 and the long IDR of TCP22 for the dynamic regulation of condensates. We added the new figure and description to support this hypothesis. “We truncated the IDR³² (Intrinsic Disorder Region) domain of CRY2 (CRY2 ^{\$\Delta\$ IDR}) and compared the dynamic responses of CRY2 ^{\$\Delta\$ IDR}-TCP22 condensates with those of CRY2-TCP22 condensates. Without the short IDR of CRY2, the TCP22- CRY2 ^{\$\Delta\$ IDR} condensates significantly decreased the dynamic compared with CRY2-TCP22 condensates (Supplementary

Fig. 9a, b). Additionally, since the TCP22 possesses a longer IDR domain than CRY2 (Supplementary Fig. 9e, f), the TCP22-CRY2^{ΔIDR} condensates were more dynamic than CRY2-CRY2^{ΔIDR} condensates with a shorter IDR domain (Supplementary Fig. 9c, d).”

5. “This indicates that CRY2-only condensates have liquid-like properties, but the liquid is probably viscous; it is likely on the boundary between the liquid-liquid phase and gel phase, which is possibly caused by the short IDR (Intrinsic Disorder Region) of CRY2 (Supplementary Fig. 5c).” Again, no data to support this claim. “CRY2 alone is sufficient to form the condensates in cells after blue light irradiation (Supplementary Fig. 4a).” This conclusion cannot be drawn, as the experiment was done in plant cells with a host of various proteins present.

RESPONSE: We agree. We revised the text to correct this conclusion: CRY2 could absorb blue light and transfer into a photo-excited state in blue light. Since CRY2^{D387A} or CRY2 in dark didn't form the condensates, it exhibited that blue light irradiation is important for CRY2 condensates formation. As shown in Supplementary Fig. 7b-d, in plant cells, CRY2-GFP condensates were found to recover after FRAP, to fuse with each other under blue light irradiation, and to be sensitive to 1, 6-hexanediol.”

- I don't understand why the authors discuss their data in the Introduction. All data should be included in results.

RESPONSE: We agree. We moved the related data into results and revised the text by: “Before the analysis on the LLPS properties of CRY2-TCP22 photobodies, we first confirmed the mass spec results that CRY2 consistently interacted with TCP22 in dark and blue light (Fig. 1a). As shown in Fig.1 b-c, the yeast two-hybrid and co-immunoprecipitated results further suggested that the physical interaction of CRY2-TCP22 was nearly constant. However, the BiFC assays indicated that in darkness, nYFP-CRY2 interacted with cYFP-TCP22 and was sufficient to reconstitute the fluorescent signal. In response to blue light, the interaction between nYFP-CRY2 and cYFP-TCP22 of the similar amount not only generated the fluorescent signal, but further formed fluorescent photobodies (Fig. 1d upper panel and Supplementary Fig. 1b). These results shed a light on the mechanism that CRY2 contacted with TCP22 via a blue light specific phase separation manner but not the blue light specific interaction.”

- Many experiments are not described in sufficient detail in the text or figure legend to figure out exactly what was done. More detail needs to be added.

RESPONSE: we have revised the figure legend part and added more descriptions, especially for the explanation of experiments. See detailed explanations in the new figure legend part.

- The CRY2 D387A mutation was not explained until well after the first experiments using this variant.

RESPONSE: We added the sentences to explain CRY2^{D387A} after the first mention. “CRY2^{D387A} is a chromophore-deficient mutant in which the residue aspartic acid is changed to alanine at position 387. Since the residue D387 of CRY2 is part of FAD-binding pocket absorbing blue light, CRY2^{D387A} does not contain the flavin which is “blind” to blue light²⁸.”

References

Arribas-Hernández, L. et al. Principles of mRNA targeting and regulation via the Arabidopsis m⁶A-binding proteins ECT2 and ECT3. Preprint at <https://www.biorxiv.org/content/10.1101/2021.04.18.440342v2> (2021).

Cho, H.S. et al. DNA gyrase is involved in chloroplast nucleoid partitioning. *The Plant cell* 16, 2665-2682 (2004).

Devlin, P.F. & Kay, S.A. Cryptochromes are required for phytochrome signaling to the circadian clock but not for rhythmicity. *The Plant cell* 12, 2499-2510 (2000).

Fang, X. et al. Arabidopsis FLL2 promotes liquid-liquid phase separation of polyadenylation complexes. *Nature* 569, 265-269 (2019).

Jung, J.H., Barbosa, A.D., Hutin, S., Kumita, J.R., Gao, M., Derwort, D., Silva, C.S., Lai, X., Pierre, E., Geng, F., et al. (2020). A prion-like domain in ELF3 functions as a thermosensor in Arabidopsis. *Nature* 585, 256-260.

Laufer, J.M. et al. Chemokine Receptor CCR7 Triggers an Endomembrane Signaling Complex for Spatial Rac Activation. *Cell reports* 29, 995-1009.e1006 (2019).

Liu, B., Zuo, Z., Liu, H., Liu, X. & Lin, C. Arabidopsis cryptochrome 1 interacts with SPA1 to suppress COP1 activity in response to blue light. *Genes & development* 25, 1029-1034 (2011).

Liu, H. et al. Photoexcited CRY2 interacts with CIB1 to regulate transcription and floral initiation in Arabidopsis. *Science* 322, 1535-1539 (2008).

- Liu, Q. et al. Molecular basis for blue light-dependent phosphorylation of Arabidopsis cryptochrome 2. *Nature communications* 8, 15234 (2017).
- Sanchez, S.E., Rognone, M.L. & Kay, S.A. Light Perception: A Matter of Time. *Molecular plant* 13, 363-385 (2020).
- Somers, D.E., Devlin, P.F. & Kay, S.A. Phytochromes and cryptochromes in the entrainment of the Arabidopsis circadian clock. *Science (New York, N.Y.)* 282, 1488-1490 (1998).
- Van Buskirk, E.K., Decker, P.V. & Chen, M. Photobodies in light signaling. *Plant physiology* 158, 52-60 (2012).
- Wang, Q. et al. Photoactivation and inactivation of Arabidopsis cryptochrome 2. *Science (New York, N.Y.)* 354, 343-347 (2016).
- Wu, J.F. et al. LWD-TCP complex activates the morning gene CCA1 in Arabidopsis. *Nature communications* 7, 13181 (2016).
- Xu, C. et al. A PIP(2)-derived amplification loop fuels the sustained initiation of B cell activation. *Science immunology* 2 (2017).
- Zavaliev, R., Mohan, R., Chen, T. & Dong, X. Formation of NPR1 Condensates Promotes Cell Survival during the Plant Immune Response. *Cell* 182, 1093-1108.e1018 (2020).
- Zuo, Z., Liu, H., Liu, B., Liu, X. & Lin, C. Blue light-dependent interaction of CRY2 with SPA1 regulates COP1 activity and floral initiation in Arabidopsis. *Current biology : CB* 21, 841-847 (2011).

REVIEWER COMMENTS

Reviewer #2 (Remarks to the Author):

It looks like the authors put in a lot of work to address the reviewer's comments. However, additional work should be done for this paper to be considered for publication.

76-78: "To address whether CRY2-TCP22 photobodies exhibit LLPS properties in plant cells, we assessed the dynamicity of CRY2-TCP22 photobodies using Fluorescence Recovery After Photobleaching (FRAP)."

I think the fusion experiments are more telling of LLPS than the FRAP (see comments about FRAP experiments below). Can you please make this statement more broad by saying, "To address whether CRY2-TCP22 photobodies exhibit LLPS properties in plant cells, we assessed the dynamicity of CRY2-TCP22 photobodies using Fluorescence Recovery After Photobleaching (FRAP) and time-lapse imaging" .

152-153: "Given longer blue-light irradiation, the CRY2-only condensates exhibited a slower recovery rate (Fig. 1g, h)"

I do not think this conclusion can be made from the FRAP experiments. To really test changes in recovery rate between different genotypes, one should bleach a portion of the target (i.e. half of your condensate) and then ask how fast it takes for the other half to recover (recovery = $t^{1/2}$). Since you bleached the entire structure, you are instead asking how fast non-bleached molecules from outside of the condensate are being incorporated into the bleached condensate, which can still provide insight into changes of material properties. A good rationale for this is presented in the text insert below which is from "Considerations and challenges in studying liquid-liquid phase separation and biomolecular condensates" by Alberti et al. 2019.

"Fluorescence recovery after photobleaching (FRAP) is often also performed on droplets as an assessment of their liquidity, and different components can vary substantially between one another in the rate of recovery due to different mobilities. This difference is especially pronounced when comparing proteins and RNAs, in which the latter appear relatively less mobile. While a highly accessible technique, there are various limitations and challenges in analysis of FRAP data. One commonly made mistake is to assume that the recovery rate of a full-FRAP reports only on the exchange rate between the dilute and the dense phase. However, the FRAP recovery rate also depends on other parameters such as the size of the photobleached droplet, mobility within a droplet and size of bleach areas and these are often not taken into account. However, it can still be useful for assessing extremes in material state or changes in the material state through time. If the structure is large enough, a half-FRAP can also be performed in addition to a full-FRAP (Brangwynne et al., 2009). In a half-FRAP, the re-arrangement of fluorescence from the bleached to the unbleached area gives more direct information on the internal mobility of molecules within a given structure. Importantly, FRAP can also be useful in assessing if the droplets are spatially homogeneous based on the pattern of recovery. As noted below, FRAP should not be used as a definitive diagnostic for determining if LLPS is the mechanism of assembly of a structure."

Line 135: The authors use the word "demonstrate" please have them tone down the language as this is still mostly shown in protoplasts and their FRAP data. Maybe change to "suggests" instead.

141-148: "Furthermore, this explains why CRY2 is different from TCP22 and other LLPS proteins, which could form the condensates alone on induction with blue light. At this stage, we hypothesized the mechanism of CRY2 condensate formation in plants as follows: after the chromophore of CRY2

senses the light signal, CRY2 undergoes oligomerization and forms the corelets (core scaffold to promote droplets) of CRY2- dependent condensates; subsequently, CRY2 recruits its interacting proteins (e.g. TCP22) and drives these proteins into phase-separation in a condensed state in plant cells.”

This seems to be contradictory to what you show in Sup. Fig 10, where CRY2 does not seem to form photobodies in the absence of TCP22 in seedlings. I think the authors need to still be very cautious here about the dosage of their protein and how much they can interpret their CRY2 solo results. In general, even if using the native promoter, you will often get multiple copies of your transgene when using protoplasts, resulting in protein OX. Since there is a mismatch between their protoplast and stable transgenic experiments, I think the authors really need to address this potential concern in the discussion and tone down some of their interpretations.

For lines 150-154, I think the reviewers deleted their hypothesis of what they think might be occurring here or forgot to motivate why they think the disordered regions might be causing this change, such as changing multi-valency or etc?

Interpretations for 150-175: I think the authors need to tone down their interpretations for supplemental figure 9. For instance, in the text (159-162) the authors write, “Additionally, since the TCP22 possesses a longer IDR domain than CRY2 (Supplementary Fig. 9e-f), the TCP22- CRY2 Δ IDR condensates were more dynamic than CRY2- CRY2 Δ IDR condensates with a shorter ID domain (Supplementary Fig. 9c-d).” However, their experiments are not directly testing length of IDR or any specific properties of the IDR other than their presence or absence. Additionally, these experiments do not directly test that the IDR provide TCP22 and CRY2’s association. Instead these results could be interpreted one of two ways, either that the IDRs may help change the 1) material properties, possibly by helping to establish a low enough multivalency so that these proteins are not permanently bound to each other in an aggregated/gel-like state or 2) change the off-rate (Kd) of the molecules, in which a faster recovery suggests a faster Kd and a slower recovery suggests a slow Kd.

Reviewer #3 (Remarks to the Author):

I thank the authors for addressing my concerns and providing additional data to support their claims. These have improved the manuscript, but I still have a few lingering concerns:

1- I think the authors misunderstood my concern with regards to use of split fluorescence proteins in most of their assays. Phase separation is driven by multivalent interactions. Providing an additional interaction interface between these two proteins (with split FPs) alters the number of available interaction interfaces between these proteins in a way that using two distinct tags does not. Thus, most of their data have some degree of artefact from using this split tag. The authors have provided other evidence, such as immunostaining of native protein, that leads me to believe that the overall message from their work is correct; however, concentrations of protein and time of blue light illumination necessary to promote condensation will, by definition, be affected by addition this additional interface between the two proteins. This caveat needs to be explicitly addressed in the text and discussion.

2- The authors have added a new line of text that is not supported by their data:

Line 159: "Additionally, since the TCP22 possesses a longer IDR domain than CRY2 (Supplementary Fig. 9e-f), the TCP22- CRY2 Δ IDR condensates were more dynamic than CRY2- CRY2 Δ IDR condensates with a shorter IDR domain (Supplementary Fig. 9c-d)."

There is no data to support that a longer IDR of TCP22 alters dynamic properties of these condensates. IDRs can either promote or suppress phase separation, depending on their biophysical characteristics. This statement needs to be removed, as there is no experimental support for it.

Minor comment:

Line 371: resent should be recent.

REVIEWER COMMENTS

Reviewer #2 (Remarks to the Author):

It looks like the authors put in a lot of work to address the reviewer's comments. However, additional work should be done for this paper to be considered for publication.

76-78: "To address whether CRY2-TCP22 photobodies exhibit LLPS properties in plant cells, we assessed the dynamicity of CRY2-TCP22 photobodies using Fluorescence Recovery After Photobleaching (FRAP)."

I think the fusion experiments are more telling of LLPS than the FRAP (see comments about FRAP experiments below). Can you please make this statement more broad by saying, "To address whether CRY2-TCP22 photobodies exhibit LLPS properties in plant cells, we assessed the dynamicity of CRY2-TCP22 photobodies using Fluorescence Recovery After Photobleaching (FRAP) and time-lapse imaging" .

152-153: "Given longer blue-light irradiation, the CRY2-only condensates exhibited a slower recovery rate (Fig. 1g, h)"

I do not think this conclusion can be made from the FRAP experiments. To really test changes in recovery rate between different genotypes, one should bleach a portion of the target (i.e. half of your condensate) and then ask how fast it takes for the other half to recover (recovery = $t^{1/2}$). Since you bleached the entire structure, you are instead asking how fast non-bleached molecules from outside of the condensate are being incorporated into the bleached condensate, which can still provide insight into changes of material properties. A good rationale for this is presented in the text insert below which is from "Considerations and challenges in studying liquid-liquid phase separation and biomolecular condensates" by Alberti et al. 2019.

"Fluorescence recovery after photobleaching (FRAP) is often also performed on droplets as an assessment of their liquidity, and different components can vary substantially between one another in the rate of recovery due to different mobilities. This difference is especially pronounced when comparing proteins and RNAs, in which the latter appear relatively less mobile. While a highly accessible technique, there are various limitations and challenges in analysis of FRAP data. One commonly made mistake is to assume that the recovery rate of a full-FRAP reports only on the exchange rate between the dilute and the dense phase. However, the FRAP recovery rate also depends on other parameters such as the size of the photobleached droplet, mobility within a droplet and size of bleach areas and these are often not taken into account. However, it can still be useful for assessing extremes in material state or changes in the material state through time. If the structure is large enough, a half-FRAP can also be performed in addition to a full-FRAP (Brangwynne et al., 2009). In a half-FRAP, the re-arrangement of fluorescence from the bleached to the unbleached area gives more direct information on the internal mobility of molecules within a given structure. Importantly, FRAP can also be useful in assessing if the droplets are spatially homogeneous based on the pattern

of recovery. As noted below, FRAP should not be used as a definitive diagnostic for determining if LLPS is the mechanism of assembly of a structure.”

Line 135: The authors use the word “demonstrate” please have them tone down the language as this is still mostly shown in protoplasts and their FRAP data. Maybe change to “suggests” instead.

141-148: “Furthermore, this explains why CRY2 is different from TCP22 and other LLPS proteins, which could form the condensates alone on induction with blue light. At this stage, we hypothesized the mechanism of CRY2 condensate formation in plants as follows: after the chromophore of CRY2 senses the light signal, CRY2 undergoes oligomerization and forms the corelets (core scaffold to promote droplets) of CRY2- dependent condensates; subsequently, CRY2 recruits its interacting proteins (e.g. TCP22) and drives these proteins into phase-separation in a condensed state in plant cells.”

This seems to be contradictory to what you show in Sup. Fig 10, where CRY2 does not seem to form photobodies in the absence of TCP22 in seedlings. I think the authors need to still be very cautious here about the dosage of their protein and how much they can interpret their CRY2 solo results. In general, even if using the native promoter, you will often get multiple copies of your transgene when using protoplasts, resulting in protein OX. Since there is a mismatch between their protoplast and stable transgenic experiments, I think the authors really need to address this potential concern in the discussion and tone down some of their interpretations.

For lines 150-154, I think the reviewers deleted their hypothesis of what they think might be occurring here or forgot to motivate why they think the disordered regions might be causing this change, such as changing multi-valency or etc?

Interpretations for 150-175: I think the authors need to tone down their interpretations for supplemental figure 9. For instance, in the text (159-162) the authors write, “Additionally, since the TCP22 possesses a longer IDR domain than CRY2 (Supplementary Fig. 9e-f), the TCP22-CRY2 Δ IDR condensates were more dynamic than CRY2- CRY2 Δ IDR condensates with a shorter ID domain (Supplementary Fig. 9c-d).” However, their experiments are not directly testing length of IDR or any specific properties of the IDR other than their presence or absence. Additionally, these experiments do not directly test that the IDR provide TCP22 and CRY2’s association. Instead these results could be interpreted one of two ways, either that the IDRs may help change the 1) material properties, possibly by helping to establish a low enough multivalency so that these proteins are not permanently bound to each other in an aggregated/gel-like state or 2) change the off-rate (Kd) of the molecules, in which a faster recovery suggests a faster Kd and a slower recovery suggests a slow Kd.

Reviewer #3 (Remarks to the Author):

I thank the authors for addressing my concerns and providing additional data to support their claims. These have improved the manuscript, but I still have a few lingering concerns:

1- I think the authors misunderstood my concern with regards to use of split fluorescence proteins in most of their assays. Phase separation is driven by multivalent interactions. Providing an additional interaction interface between these two proteins (with split FPs) alters the number of available interaction interfaces between these proteins in a way that using two distinct tags does not. Thus, most of their data have some degree of artefact from using this split tag. The authors have provided other evidence, such as immunostaining of native protein, that leads me to believe that the overall message from their work is correct; however, concentrations of protein and time of blue light illumination necessary to promote condensation will, by definition, be affected by addition this additional interface between the two proteins. This caveat needs to be explicitly addressed in the text and discussion.

2- The authors have added a new line of text that is not supported by their data:

Line 159: “Additionally, since the TCP22 possesses a longer IDR domain than CRY2 (Supplementary Fig. 9e-f), the TCP22- CRY2 Δ IDR condensates were more dynamic than CRY2- CRY2 Δ IDR condensates with a shorter IDR domain (Supplementary Fig. 9c-d).”

There is no data to support that a longer IDR of TCP22 alters dynamic properties of these condensates. IDRs can either promote or suppress phase separation, depending on their biophysical characteristics. This statement needs to be removed, as there is no experimental support for it.

Minor comment:

Line 371: resent should be recent.

Responses to the reviewers' comments

Notes:

The reviewers' comments are underlined, and the authors' revised text and newly added figures are highlighted by the blue color, which are also searchable in the revised manuscript to find their locations in the revision.

Responses to Review #2

1. I think the fusion experiments are more telling of LLPS than the FRAP (see comments about FRAP experiments below). Can you please make this statement more broad by saying, "To address whether CRY2-TCP22 photobodies exhibit LLPS properties in plant cells, we assessed the dynamicity of CRY2-TCP22 photobodies using Fluorescence Recovery After Photobleaching (FRAP) and time-lapse imaging".

RESPONSE: We agree. Indeed, the time-lapse imaging of fusion assay is also important for the LLPS properties of CRY2-TCP22 photobodies. We really thank for the reviewer's careful consideration of our manuscript, we also appreciate the reviewer's comment on our manuscript in a rigorous manner, which indeed improved our story. We revised the previous sentence as: "To address whether CRY2-TCP22 photobodies exhibit LLPS properties in plant cells, we assessed the dynamicity of CRY2-TCP22 photobodies using Fluorescence Recovery After Photobleaching (FRAP) and time-lapse imaging^{23, 24}"

2. I do not think this conclusion can be made from the FRAP experiments. To really test changes in recovery rate between different genotypes, one should bleach a portion of the target (i.e. half of your condensate) and then ask how fast it takes for the other half to recover (recovery = $t^{1/2}$). Since you bleached the entire structure, you are instead asking how fast non-bleached molecules from outside of the condensate are being incorporated into the bleached condensate, which can still provide insight into changes of material properties. A good rationale for this is presented in the text insert below which is from "Considerations and challenges in studying liquid-liquid phase separation and biomolecular condensates" by Alberti et al. 2019.

RESPONSE: We thank the reviewer for raising this rigorous issue of CRY2-TCP22 LLPS. Actually, we have considered the half-FRAP assay before. However, we were really irresolute previously whether we should add the half-FRAP to further exhibit the recovery rate of those LLPS condensates or not, because we have confirmed the LLPS properties with many ways including: in vitro and in vivo, in protoplasts and in stable lines or WT, by the fluorescence protein and by the immunostaining assay, and our manuscript embraced more LLPS-related technique than most LLPS studies. To avoid our manuscript becoming potential redundancy, we ignored the half-FRAP assay in last revision. On the other hand, we agree that the half-FRAP provides the additional information on the internal mobility of molecules in LLPS condensates. Here, we really thank the reviewer for this suggestion. We further confirmed our

conclusion as: “Given longer blue-light irradiation, the CRY2-only condensates exhibited a slower recovery rate (Fig. 1g, h)”, and we added the half-FRAP assay in this revision. As shown in the newly added Supplementary Fig. 9e, the CRY2-only condensates exhibited weaker internal mobility than that of CRY2-TCP2 condensates, which further confirmed our result in our manuscript.

3. Line 135: The authors use the word “demonstrate” please have them tone down the language as this is still mostly shown in protoplasts and their FRAP data. Maybe change to “suggests” instead.

RESPONSE: We agree, we have revised the text: “These results suggested that CRY2 photobodies reported previously^{15, 29, 30}, are actually CRY2 intrinsic liquid-like condensates in plant cells.”

4. This seems to be contradictory to what you show in Sup. Fig 10, where CRY2 does not seem to form photobodies in the absence of TCP22 in seedlings. I think the authors need to still be very cautious here about the dosage of their protein and how much they can interpret their CRY2 solo results. In general, even if using the native promoter, you will often get multiple copies of your transgene when using protoplasts, resulting in protein OX. Since there is a mismatch between their protoplast and stable transgenic experiments, I think the authors really need to address this potential concern in the discussion and tone down some of their interpretations.

RESPONSE: This is the new issue raised by last revision which should be revised last time. And we thank the reviewer for pointing out this relevant issue that we had overlooked in last revision. we revised the text as: “Furthermore, this explains why CRY2 is different from TCP22 and other LLPS proteins²⁵⁻²⁷, which forms the condensates under blue light and in an oligomer dependent manner. At this stage, we hypothesized the mechanism of CRY2 condensate formation in plants as follows:” The reviewer raised the interesting scientific questions: since CRY2 interacts with multiple proteins including SPAs, PIFs, CIB1, TCP22, *etc.*, how do those proteins affect the formation of CRY2 LLPS, which one is the most important to regulate CRY2 LLPS, and even whether the formation of CRY2 condensates depends on those proteins? We will follow up and investigate these questions in the future study.

We further thank the reviewer for raising the native promoter issue. In this revision, we checked whether the native promoter causes protein OX to affect the morphology of CRY2 related condensates in our study. As shown in the newly added Supplementary Fig. 3d, the CRY2 condensates exhibited the similar status in WT and *ProTCP22::Myc-TCP22/tcp20tcp22* plant with immunostaining assay. The CRY2 condensates in both WT and *ProTCP22::Myc-TCP22/tcp20tcp22* plant were further inhibited by 1,6 Hexanodiol (Supplementary Fig. 3d).

5. For lines 150-154, I think the reviewers deleted their hypothesis of what they think might be occurring here or forgot to motivate why they think the disordered regions might be causing this change, such as changing multi-valency or etc?

RESPONSE: we didn’t delete or forget it. We added the truncated assay as the suggestion of reviewers in last revision, so it came to the lines 170-175.

6. Interpretations for 150-175: I think the authors need to tone down their interpretations for supplemental figure 9. For instance, in the text (159-162) the authors write, “Additionally, since the TCP22 possesses a longer IDR domain than CRY2 (Supplementary Fig. 9e-f), the TCP22-CRY2 Δ IDR condensates were more dynamic than CRY2- CRY2 Δ IDR condensates with a shorter ID domain (Supplementary Fig. 9c-d).” However, their experiments are not directly testing length of IDR or any specific properties of the IDR other than their presence or absence. Additionally, these experiments do not directly test that the IDR provide TCP22 and CRY2’ s association. Instead these results could be interpreted one of two ways, either that the IDRs may help change the 1) material properties, possibly by helping to establish a low enough multivalency so that these proteins are not permanently bound to each other in an aggregated/gel-like state or 2) change the off-rate (Kd) of the molecules, in which a faster recovery suggests a faster Kd and a slower recovery suggests a slow Kd.

RESPONSE: We agree. Compared to the half-FRAP assay (as discussed previously), the experiment exhibiting in supplementary Fig. 9c-d did not directly explain the internal mobility and the recovery rate of CRY2 related condensates. So we added the half-FRAP assay to replace the experiment of supplementary Fig. 9c-d in last revision. We also revised the text: “Additionally, we further examined the internal mobility and recovery rate of CRY2-TCP22 and CRY2-only condensates with half-FRAP assay. As shown in supplementary Fig. 9e, the half-bleached CRY2-TCP22 condensates exhibited a faster recovery rate compared with CRY2-only condensates.”

Responses to Review #3

1. I think the authors misunderstood my concern with regards to use of split fluorescence proteins in most of their assays. Phase separation is driven by multivalent interactions. Providing an additional interaction interface between these two proteins (with split FPs) alters the number of available interaction interfaces between these proteins in a way that using two distinct tags does not. Thus, most of their data have some degree of artefact from using this split tag. The authors have provided other evidence, such as immunostaining of native protein, that leads me to believe that the overall message from their work is correct; however, concentrations of protein and time of blue light illumination necessary to promote condensation will, by definition, be affected by addition this additional interface between the two proteins. This caveat needs to be explicitly addressed in the text and discussion.

RESPONSE: We really thank for the patient explanation of the reviewer, and we also thank for the suggestion of the reviewer. we agree with the issue of fluorescence proteins, and we added these sentences to caution: Sentence 1: “Since the tags of the split fluorescence protein may potentially affect the interaction interface of proteins and the overexpression may affect the protein concentration to promote condensation, we further confirmed that the native CRY2 formed the condensates in the wild type plant cells utilizing the immunostaining assay (Supplementary Fig. 3a-b); and CRY2-TCP22 also formed the condensates in a blue light dependent manner when CRY2 and TCP22 expressed under

their native promoters, respectively (Supplementary Fig. 3c) ”; Sentence 2: “To avoid the potential interaction interference of the split fluorescence protein, we further confirmed this suggestion by the immunostaining assay, in which CRY2 only formed the condensates with Myc-TCP22 driven by the native promoter in blue light, in contrast, no condensate was exhibited in *tcp20tcp22* mutant or in red light (Supplementary Fig.10 a, b).” Sentence 3 in discussion: “In any case, our study confirmed the blue light specific formation of CRY2-related LLPS utilizing the fluorescence protein, immunostaining and *in-vitro* reconstitution assay; and it shed light on the mechanism of LLPS-based light input to the circadian clock.”

2. The authors have added a new line of text that is not supported by their data:

Line 159: “ Additionally, since the TCP22 possesses a longer IDR domain than CRY2 (Supplementary Fig. 9e-f), the TCP22- CRY2 Δ IDR condensates were more dynamic than CRY2- CRY2 Δ IDR condensates with a shorter IDR domain (Supplementary Fig. 9c-d).”

There is no data to support that a longer IDR of TCP22 alters dynamic properties of these condensates. IDRs can either promote or suppress phase separation, depending on their biophysical characteristics. This statement needs to be removed, as there is no experimental support for it.

RESPONSE: We agree. We deleted this sentence. Actually, the reviewer 2# also raised the similar issue, and we added half-FRAP assay to further confirm the dynamic property issue.

3. Minor comment:

Line 371: resent should be recent.

RESPONSE: Thanks, and we have revised this wrong spelling.

REVIEWER COMMENTS

Reviewer #2 (Remarks to the Author):

Major comment:

I think the authors misunderstood a previous comment of mine in the last round of revisions. I am asking whether CRY2 bodies require TCP22 for their formation? lines 114-184 suggest that CRY2 can form bodies in the absence of TCP22 based on their protoplast system. However, their immunostaining data in supplemental figure 10 suggests that TCP22 is required. Thus, does that mean that CRY2-only condensates are an artifact generated by their protoplast system? This needs to be addressed.

Additionally, the authors put forth a model in lines 148-155, but I think it needs to be revised to saying something more along the lines of, "After the chromophore of CRY2 senses light, CRY2 will undergo oligomerization, where it will co-condense with TCP22 and possibly other interactors." I think this is more appropriate than saying that CRY2 forms a correlate and then recruits TCP22 because (1) they have not tested nor shown that CRY2 forms a correlate, (2) TCP22 seems to be required for CRY2 bodies, and (3) TCP22 and CRY2 are undergoing continuous interactions.

Minor changes:

Line 82: should be "hydrophobic interactions" vs "hydrophobic interaction"

Supplemental figure 1B Y-axis should be "NFI" assuming the authors mean normalized fluorescent intensity.

Line 132-134: the writing is awkward. Maybe, " Similar to TCP22, reconstituted CRY2 formed droplets in vitro that underwent fusion and became inhibited by 1,6-hexanediol."

Reviewer #3 (Remarks to the Author):

I have no further suggestions to improve this manuscript.

Responses to the reviewers' comments

Notes:

The reviewers' comments are underlined, and the authors' revised text are highlighted by the blue color, which are also searchable in the revised manuscript to find their locations in the revision.

Responses to Review #2

1. I think the authors misunderstood a previous comment of mine in the last round of revisions. I am asking whether CRY2 bodies require TCP22 for their formation? lines 114-184 suggest that CRY2 can form bodies in the absence of TCP22 based on their protoplast system. However, their immuno-staining data in supplemental figure 10 suggests that TCP22 is required. Thus, does that mean that CRY2-only condensates are an artifact generated by their protoplast system? This needs to be addressed.

Responses: Thus, does that mean that CRY2-only condensates are an artifact generated by their protoplast system? CRY2-only condensates are not the artifact in protoplast, because we have analyzed the CRY2-only condensates in Col-4 background protoplast. Unlike that in CRY2/TCP22 overexpression background, the expression level of TCP22 is not enough to support the fast formation and the bigger size of CRY2-condensates in Col-4 background, but TCP22 still expressed in Col-4 background, compared with that there is no TCP22 in *tcp20/tcp22* background (supplemental figure 10 of last revision). And in supplemental figure 10 of last revision, the immuno-staining assay utilized the *tcp20/tcp22* plants, which has no TCP22 and we didn't observe the CRY2-condensate. Furthermore, we could also observe the CRY2 condensates in wild-type (Col-4) cells without the additional expression of any protein by immuno-staining assay, which could further confirm our CRY2-only condensates assay with the protoplast system.

"I am asking whether CRY2 bodies require TCP22 for their formation?" As we responded in last round revision, we agreed it is a good and interesting question, but we can not make a conclusion in this study. Because: (1) we have only one kind of evidence (only in immuno-staining assay) to show CRY2 need TCP22 to form the condensates. It is not like the issue of TCP22 forming the condensates depending on CRY2, which we have protoplast evidence, immuno-staining evidence, *cry1cry2* mutant background assays evidence and the CRY2^{D387A} negative evidence, etc. (2) After several rounds of revision, we have added a lot of assays in this manuscript which forms such a complicated story that the editor re-organized our story, so there is no more space for adding new assays to prove this issue. Additionally, this issue is not the main one in this study. But we thank the reviewer for raising this issue, and it's really a nice scientific issue for future study which mainly investigates how the CRY2-interacting proteins, including SPAs, PIFs, CIBs, PPKs, TCP22, etc., affect the formation of CRY2 condensates; which one is the most important to regulate CRY2 LLPS; and how the formation of CRY2 condensates depends on those proteins.

2. Additionally, the authors put forth a model in lines 148-155, but I think it needs to be revised

to saying something more along the lines of, "After the chromophore of CRY2 senses light, CRY2 will undergo oligomerization, where it will co-condense with TCP22 and possibly other interactors." I think this is more appropriate than saying that CRY2 forms a correlate and then recruits TCP22 because (1) they have not tested nor shown that CRY2 forms a correlate, (2) TCP22 seems to be required for CRY2 bodies, and (3) TCP22 and CRY2 are undergoing continuous interactions.

Responses: We thank and prefer the description of reviewers. Since we re-organized the manuscript as the suggestion of the editor, this sentence did not exist in the revised manuscript.

Minor changes:

1. Line 82: should be "hydrophobic interactions" vs "hydrophobic interaction"

Responses: We thank the reviewer, and have changed "hydrophobic interaction" to "hydrophobic interactions"

2. Supplemental figure 1B Y-axis should be "NFI" assuming the authors mean normalized fluorescent intensity.

Responses: We agree, we have changed "MFI" to "NFI"

3. Line 132-134: the writing is awkward. Maybe, " Similar to TCP22, reconstituted CRY2 formed droplets in vitro that underwent fusion and became inhibited by 1,6-hexanediol."

Responses: We thank for the suggestion of the reviewer, and this sentence has been re-organized, so it was not in the revised manuscript.